# Neuronal tuning and population representations of shape and category in human visual cortex

Vasiliki Bougou [1,2], Michaël Vanhoyland [1,2,3], Alexander Bertrand [4], Wim Van Paesschen [5,6], Hans Op De Beeck [7], Peter Janssen [2] ✉ & Tom Theys [1,3]

Object recognition and categorization are essential cognitive processes which engage considerable neural resources in the human ventral visual stream. However, the tuning properties of human ventral stream neurons for object shape and category are virtually unknown. We performed large-scale recordings of spiking activity in human Lateral Occipital Complex in response to stimuli in which the shape dimension was dissociated from the category dimension. Consistent with studies in nonhuman primates, the neuronal representations were primarily shape-based, although we also observed category-like encoding for images of animals. Surprisingly, linear decoders could reliably classify stimulus category even in data sets that were entirely shape-based. In addition, many recording sites showed an interaction between shape and category tuning. These results represent a detailed study on shape and category coding at the neuronal level in the human ventral visual stream, furnishing essential evidence that reconciles human imaging and macaque single-cell studies.

Object recognition and categorization are fundamental cognitive processes, essential for understanding and interpreting the visual world. The lateral and ventral occipitotemporal cortices (OTC) are key regions involved in these processes[1,2] Nevertheless, the precise functional organization, neuronal tuning properties and hierarchical structure of this large cortical region remain unclear.

Functional magnetic resonance (fMRI) studies in humans have shown that the Lateral Occipital Complex (LOC) is particularly sensitive to shape features[3,4], and bears remarkable similarities with the macaque inferior temporal cortex (ITC)[5–7]. Along the hierarchical organization of the human ventral visual stream, functional activations emerge suggesting the existence of more categorical object representations for diverse stimuli, including faces[8], bodies[9], scenes[10], hands[11], letter strings[12], and food items[13,14].

However, the current body of evidence is insufficient to draw definitive conclusions regarding category selectivity at the neuronal level in the human OTC. First, prior research has tested a relatively small number of categories. Additionally, the limited spatiotemporal resolution of fMRI does not allow to make strong inferences about the underlying neuronal selectivities without a number of assumptions[15–17]. Thus, to gain a deeper understanding of the neural mechanisms underlying object processing, single-cell recordings in macaques have been crucial, a model that has been validated by evidence of a common organization of object space in humans and monkeys[18].

[1]Research Group of Experimental Neurosurgery and Neuroanatomy, Department of Neurosciences, KU Leuven and the Leuven Brain Institute, Leuven, Belgium. [2]Laboratory for Neuro—and Psychophysiology, Research Group Neurophysiology, Department of Neurosciences, KU Leuven and the Leuven Brain Institute, Leuven, Belgium. [3]Department of Neurosurgery, University Hospitals Leuven, Leuven, Belgium. [4]Department of Electrical Engineering, KU Leuven, Leuven, Belgium. [5]Department of Neurology, University Hospitals Leuven, Leuven, Belgium. [6]Laboratory for Epilepsy Research, KU Leuven, Leuven, Belgium. [7]Laboratory Biological Psychology, Department of Neurosciences, KU Leuven, Leuven, Belgium. ✉e-mail: peter.janssen@kuleuven.be

In macaques, neurons in prefrontal and posterior parietal cortex exhibit distinct categorical representations for learned categories, indicating their crucial involvement in higher-level visual processing. Conversely, the ITC shows only weak or absent category effects[19–21] (except in face or body patches[22,23]). However, in humans, an fMRI study[24] manipulated shape type and category independently, and reported both shape and category sensitivity in lateral and ventral occipitotemporal cortex, with a gradual progression from more shape-based representations posteriorly to more category-based representations in more anterior brain regions. Yet again, in the absence of data on the actual neuronal tuning properties of human visual neurons it is difficult to relate these fMRI findings on human lateral occipito-temporal cortex to the existing electrophysiological evidence in the macaque ventral visual stream.

To bridge this looming gap between human fMRI and macaque electrophysiology, we recorded multi-unit activity (MUA) and high-gamma (HG) responses in the human LOC using intracortical micro-electrode arrays during the presentation of shapes belonging to different categories, in which the shape dimension was dissociated from the category dimension as in ref. 24. Note that in this study, we use the terms 'shape' and 'category' to describe the stimulus set in line with many previous imaging and computational studies[25,26]. We employed a diverse set of analysis techniques to investigate shape and category representations both at the individual channel level and at the population level. We found mainly shape-based representations with a large number of shape-category interactions in individual recording channels. At the population level, the neuronal dissimilarities did not correlate with behavioral category judgments, but linear decoders could correctly classify category information in every array tested. These results represent a detailed study of shape–and category coding at the level of multiunit spiking activity in human visual cortex.

## Results

Three neurosurgical patients with refractory epilepsy, one of whom received two arrays, underwent implantation of in total four micro-electrode arrays (96-channel Utah arrays) during a semi-chronic setting. These arrays were positioned near subdural grids for intracranial clinical seizure monitoring. After 14 days, the arrays and grids were removed without any additional incisions. Figure 1A shows the reconstructed anatomical locations of the arrays (Montreal Neurological Institute (MNI) coordinates in Table 1). To relate our findings to previous fMRI studies[4,6], we also show the average normalized net responses of all visually-responsive channels to the intact versus scrambled stimuli (classic LOC stimuli and naturalistic LOC images). Although the degree of selectivity for image scrambling and the response latency differed between the arrays, the significantly stronger responses to intact images of objects compared to scrambled ones demonstrate that all arrays were located in shape-sensitive lateral occipital cortex, in agreement with ref. 27. Moreover, the MNI coordinates of every array were overlapping with the object versus scrambled parcels described in ref. 28 (see Table S1, Fig. S1). Figure 1B illustrates the main stimulus set presented to patients, as described in ref. 24. The patients were awake and performed either a passive fixation task (patient 1) or a variant of the same passive fixation with a distractor (patient 2, patient 3), in which the patients pressed a button with their right hand whenever a distractor (red or green cross) appeared at the fixation point, randomly in ~2% of the trials). The stimulus set, consisting of 54 grayscale images (exemplars), distinguishes between the shape and category dimensions. These images belong to one of six category groups and one of nine shape types, ensuring orthogonality between the two dimensions. Specifically, each shape type includes one exemplar from all categories, and each category encompasses one exemplar from all shape types.

### Single-channel responses reveal tuning complexity

We recorded from 237 visually responsive MUA sites (array 1: 51, array 2: 94, array 3: 27, array 4: 65) and 332 visually responsive HG sites (60–120 Hz; array 1: 85, array 2: 96, array 3: 86, array 4: 65) (we identified visually responsive sites for all conditions, not just the two most preferred ones, and found that all 96 channels in all four arrays were visually responsive at the HG level. However, after repeating the entire analysis, we obtained qualitatively the same results at both the single-channel and population levels.). First, we determined the selectivity for shape, for category and any shape-category interactions using 2-way ANOVA on the net MUA and HG responses (see Methods). Figure 2 shows the MUA (Fig. 2A, B, C) and HG (Fig. 2D, E, F) responses for six

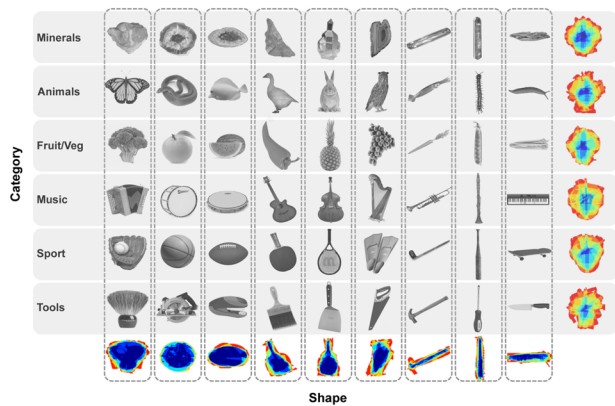

**Fig. 1 | Methods. A** Microarray recording locations plotted on a common brain, with a different number for each array. Lineplots of average normalized multi-unit activity of all visually responsive channels per array for intact (purple) and scrambled (orange) objects for the LOC-Naturalistic images (left plots) and the LOC-classic images (right plots). The stars indicate the significant ($p < 0.05$), corrected for multiple comparisons using a Tukey's test with 95% confidence interval, difference between the intact and scrambled object responses following a one-way ANOVA conducted using a sliding window approach with a window size of 100 ms and 50% overlap. Data are presented as mean values ± SEM across trials. **B** Experimental stimuli for the shape-category experiment[24]. The stimulus set consists of 6 object categories (rows) and 9 shape types (columns); 54 unique images in total. The pixelwise overlap is shown in the last row and last column and corresponds to the sum of all images from each shape type and each category type respectively.

(three MUA sites and three HG sites, which are not the same) example channels. The first example channel (recorded in array 2, Fig. 2A) responded strongly to several shape types (e.g., shape type 5, 6, and 8), but much less to other shape types (e.g., shape type 7 and 9, main effect of shape $p_{shape} = 0.0001$). The different categories within each shape type evoked similar responses in this MUA site ($p_{category} = 0.52$, $p_{interaction} = 0.65$, Supplementary Table 1 for details on statistics). The robust shape selectivity and lack of category selectivity were also evident in the average responses of the HG example site (recorded in array 2) (Fig. 2D). In contrast, the example site in Fig. 2B (recorded in array 3) responded strongly to certain exemplars of the category 'animals' (those from shape types 5 and 6), which represents a significant shape × category interaction ($p = 0.0007$) with a weak main effect of category ($p = 0.026$) and no significant main effect of shape ($p = 0.06$, Supplementary Table 1). The shape x category interaction effect was even more pronounced in the HG example site than in the MUA example site ($eta^2_{MUA} = 0.07$, $eta^2_{HG} = 0.19$, Fig. 2E, and Supplementary Table 1). Finally, the example site shown in Fig. 2C (from array 2) displayed stronger neural responses to certain members of a

particular shape type (e.g., 'Fruits' for shape type 6), which constituted another type of interaction between shape and category ($p = 0.000$), combined with a main effect of shape ($p = 0.00002$), but no significant effect of category ($p = 0.46$, Supplementary Table 1). These interactions could be due to selectivity for the specific exemplar (e.g., the fruit for shape type 6 is a bunch of grapes), to subtle differences between the members of the same shape or category in their shape and category properties, or due to variations in other dimensions such as variations in contour or texture. Raster plots and wave forms of single units are shown in Fig. S2. Overall, these results suggest that while shape selectivity is a dominant feature of the visual responses in the sites of human occipitotemporal cortex that we sampled, interactions between shape and category were also observed in a subset of neural sites. (For plots on shape selectivity across all 54 stimuli, latencies and receptive field sizes, see Supplementary Figs. S3, S4).

To illustrate the shape and category responses of all visually-responsive channels, Fig. 3A, B show an overview of the z-scored responses (see Methods) per array at the MUA and HG level, respectively. We ordered the channels from top to bottom based on their selectivity as determined in the 2-way ANOVA with factors *shape type* and *category*: channels indicated by the blue bracket showed a main effect of shape type only, channels indicated by the yellow bracket showed a main effect of category only, and channels with the green bracket showed a significant shape type x category interaction (sometimes in combination with a main effect of shape type and/or category). The channels below the green bracket were visually responsive but did not show any significant effect in the two-way ANOVA. The order of the columns (from left to right) was determined

**Table 1 | MNI coordinates of Utah arrays**

| ARRAYS | X | Y | Z |
|---|---|---|---|
| 1 | 42 | −76 | −1 |
| 2 | −35 | −89 | −8 |
| 3 | −38 | −84 | −5 |
| 4 | −41 | −83 | 9 |

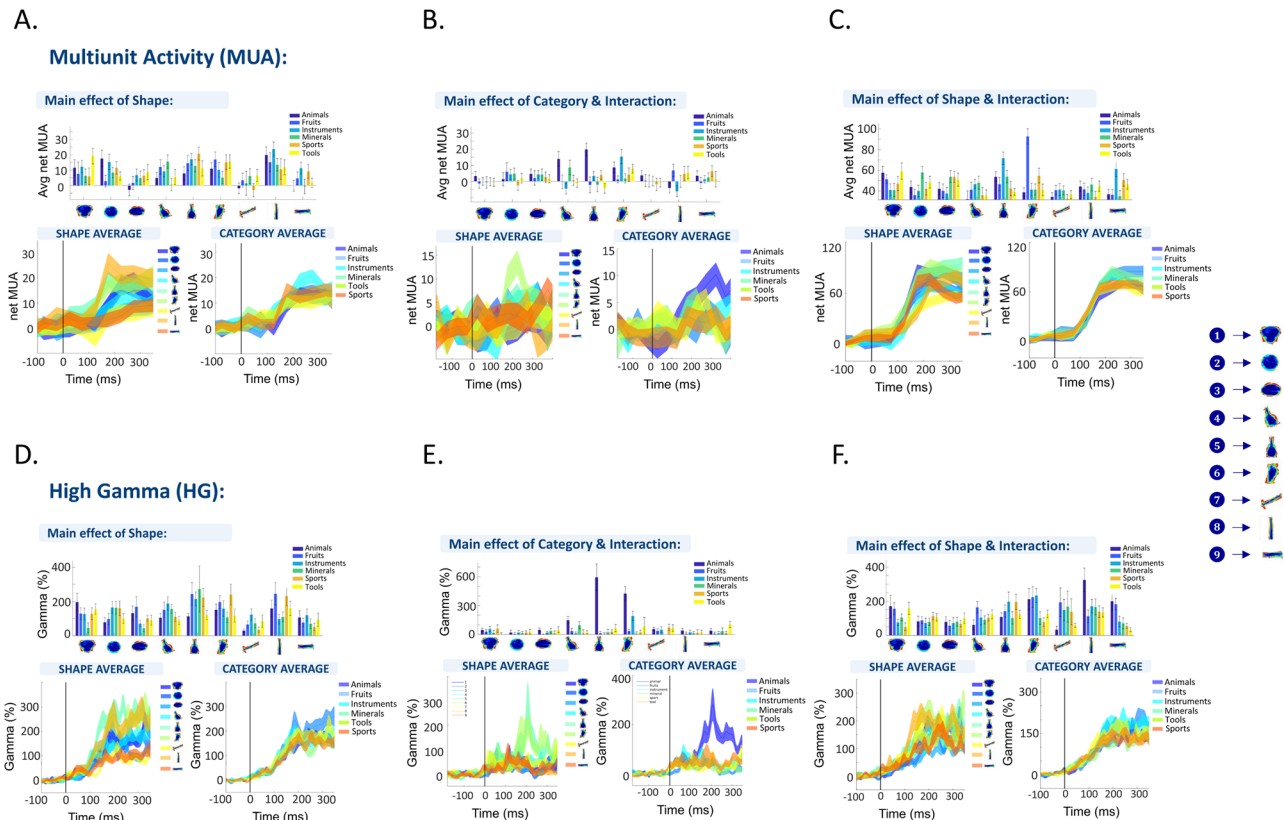

**Fig. 2 | Example sites.** Example sites for MUA (**A**, **B**, **C**) and LFP high-gamma (**D**, **E**, **F**) responses. For each channel the height of the bar indicates the average net MUA across time (channel in (**A**): 75–275 ms after stimulus onset, channel in (**B**): 125–325 ms after stimulus onset, channel in (**C**): 75–275 ms after stimulus onset) for each of the 54 stimuli, or the average normalized high-gamma activity (channel in (**A**): 25–225 ms after stimulus onset, channel in (**B**): 125–325 ms after stimulus onset, channel in (**C**): 75–275 ms after stimulus onset). The different colors correspond to the six different semantic categories and the different columns to the nine individual shape types. The error bars indicate the standard error across trials. The line plots below the bar plots show the responses over time, averaged across each shape type (left) and each category (right). The width of the line indicates the standard error across trials.

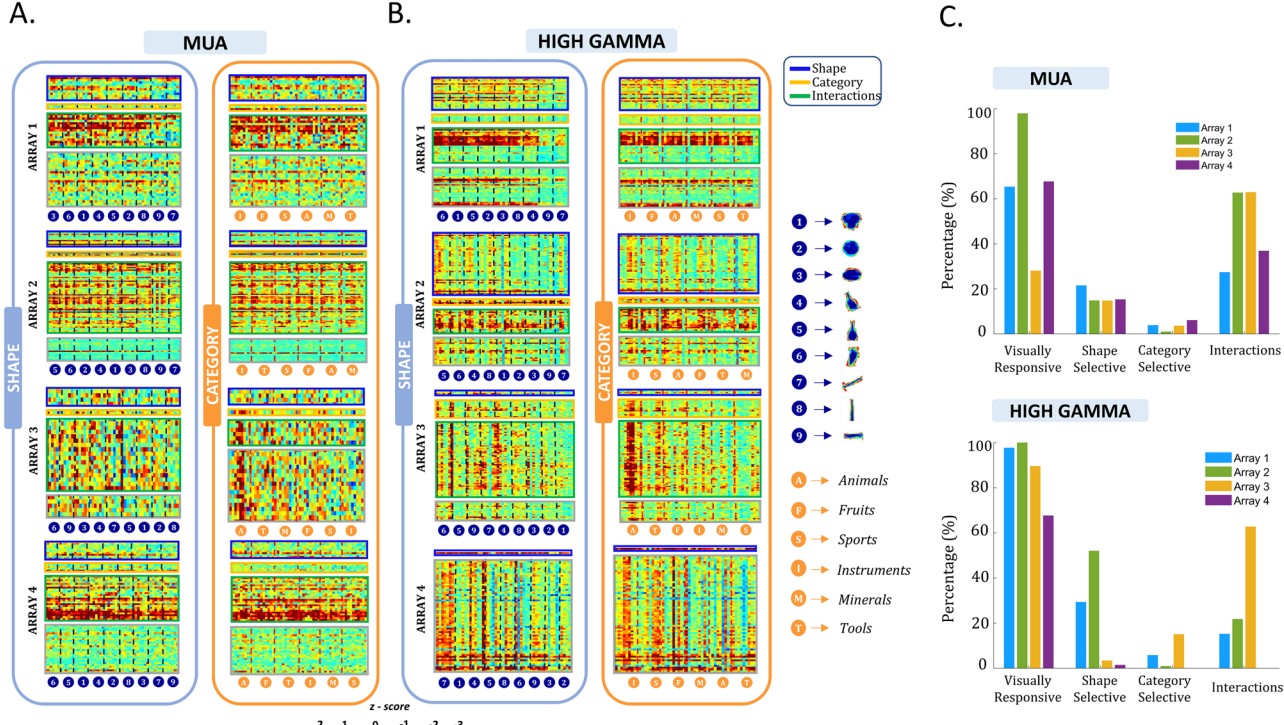

**Fig. 3 | Overview of responses for all visually responsive sites. A** Net z-scored MUA responses averaged over time (after stimulus onset) and ordered per array for all visually responsive sites. The numbers indicate the shape group and the letters the semantic category. The channels were ordered according to their selectivity which is indicated by the outline of each block (blue: significant shape main effect, orange: significant category main effect, green: significant interaction between shape and category). **B** Same plots as in (**A**), but for the normalized high-gamma power. **C** Summary of the results of the 2-way ANOVA (upper plots MUA, lower plots HG; blue: array 1, green: array 2; yellow: array 3; purple: array 4) The first column shows the percentage of visually responsive channels. The second, third, and fourth columns show the percentage of the visually responsive channels that have a significant effect of shape type, of category, and interactions respectively.

based on the average response of all visually-responsive channels across each array separately. The plots ordered according to shape type (left panels in Fig. 3A, B) clearly illustrate that our stimulus set evoked strong MUA and HG responses on a large number of recording channels. To assess the selectivity for the 54 individual stimuli, we computed the Selectivity Index (Swidth) which quantifies the extent to which a channel exhibits preference for a specific stimulus, providing a numerical measure of its tuning specificity (see Methods). This measure ranges from 0 for sites that respond equally to all stimuli to 1 for sites that only respond to one of the stimuli. It can provide a better understanding of each channel's selectivity characteristics in response to diverse stimuli. The selectivity was relatively broad for all arrays (Fig. S3), as illustrated by the $S_{width}$ index quantifying the number of stimuli evoking a response (see methods. Median $S_{width}$ MUA: $s_{array1} = 0.69$, $s_{array2} = 0.62$, $s_{array3} = 0.86$, $s_{array4} = 0.7$, median $S_{width}$ HG: $s_{array1} = 0.5$, $s_{array2} = 0.52$, $s_{array3} = 0.69$, $s_{array4} = 0.52$). In addition, we calculated the D-prime ($d'$) (1), (Fig. S3) assessing each channel's effectiveness in distinguishing the target stimulus from non-preferred stimuli (see Methods). Our findings revealed robust selectivity for individual stimuli, as evident from consistently high $d'$ values, frequently surpassing 2, across all four arrays. (Median $d'$ MUA: $d'_{array1} = 1.76$, $d'_{array2} = 2.06$, $d'_{array3} = 1.98$, $d'_{array4} = 2.73$, median $S_{width}$ HG: $d'_{array1} = 1.79$, $d'_{array2} = 1.6$, $d'_{array3} = 1.99$, $d'_{array4} = 1.35$).

Visual inspection does not suggest a clear preference for specific shape types in any of the arrays. When plotting the responses according to category (right panels in Fig. 3A, B), the results were qualitatively similar, except for the category "animals" in array 3, which clearly evoked strong responses to a subset of shape types belonging to this category, as illustrated in the example channels in Fig. 2B, D (see also Supplementary Fig. S5 for responses to animals with faces compared to animals without faces, and responses to animals from

different taxonomic classes). To investigate the overall shape type or category preference for each array more quantitatively, we averaged the MUA and HG responses across all visually-responsive channels (Fig. 4A). Arrays 1, 2, and 4 responded significantly less to shape types 7, 8, and 9 (which were characterized by a lower surface area and high aspect ratio), whereas for array 3, the MUA response to the category 'animals' was significantly higher compared to the other categories (Fig. 4A). The HG responses ranked according to shape type (Fig. 3B left panel) appeared very similar to the MUA responses, which was supported by the significant correlations between MUA and HG responses for all arrays (Fig. 4B). When plotted according to category, the high gamma responses of array 3 contained an even more pronounced preference for the category 'animals' than the MUA responses (Fig. 3B and eta² values in Fig. S6B).

Further analysis of all individual visually-responsive electrodes (using two-way ANOVA with factors *shape type* and *category*) confirmed the high diversity of neural tuning for shape type and category. At the MUA level, the highest number of channels showed a significant interaction between shape type and category for all arrays (Fig. 3C). More specifically, out of the 237 visually responsive MUA sites, 39 sites (16%) were significantly selective for the shape type dimension alone, merely 8 sites (3%) showed a significant main effect of category alone, compared to 114 sites (48%) with interactions between shape type and category (chi² = 143, $p < 0.0001$). At the HG level, we also observed mainly shape type selectivity and shape-category interactions, although Array 1 and Array 2 showed more channels with a significant main effect of shape type (chi² = 6.8, $p < 0.0001$). In two arrays, the proportion of significant shape type × category interactions was significantly higher in the MUA (27 and 63% for arrays 1 and 2, respectively) compared to the HG responses (12 and 22% for arrays 1 and 2, respectively; array 3 had a

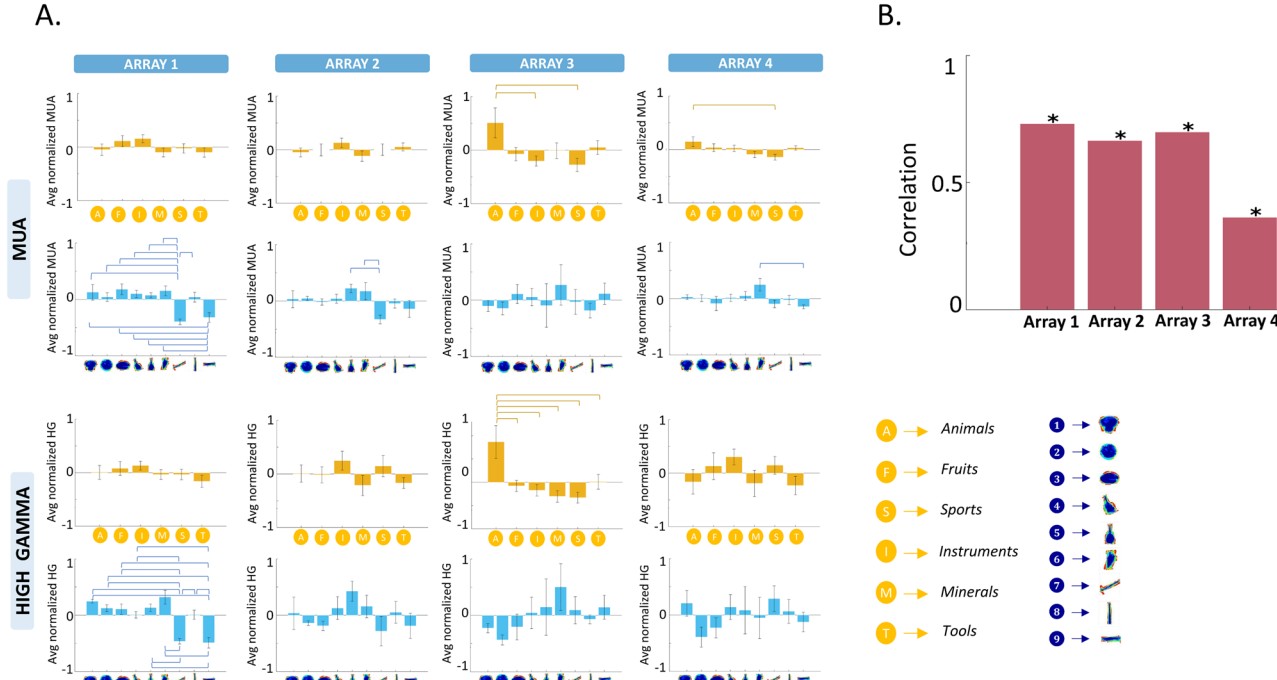

**Fig. 4 | Overview of average responses. A** Average MUA (upper panel) and high-gamma (lower panel) both across visually responsive channels and within the category (orange bars) and shape (blue bars) dimensions. The height of each bar represents the mean response, while the error bar indicates the standard error across channels. Brackets indicate significant differences between shape members or semantic categories ($n = 54$ for all arrays). **B** Spearman correlation between the MUA and the high-gamma average (across visually responsive channels) responses.

similar proportion of interactions in MUA and HG, and for array 4 the HG signal was of low quality).

To test the effect sizes for shape type and category, we compared the eta$^2$ of all sites with significant effects (Fig. S6). Overall, the eta$^2$ values for shape type were higher than for category in arrays 1, 2, and 4, and this difference in eta$^2$ was more pronounced for sites displaying a main effect of shape. Interestingly, in arrays 1, 2, and 4, even for channels with only a significant interaction or with both significant shape and category main effects, eta$^2$ was significantly stronger for shape type compared to category. However, this was not the case for the shape type × category interaction channels of array 3, where both shape and category effect sizes were similarly strong.

We also analyzed the broadband LFP responses (Evoked Response Potential, ERP) and compared shape type and category selectivity to with the MUA and HG responses. Overall, we observed fewer channels with significant selectivity in the ERP, which was primarily for shape type and only appeared in arrays 1 and 2 (Fig. S7). Moreover, a ranking analysis demonstrated that the stimulus preference was highly preserved between MUA and HG activity, but not between MUA and ERP (Fig. S8, Table S2). Therefore, all remaining analyses were performed on MUA and HG responses. Single-unit waveforms and raster plots are illustrated in Fig. S2.

### Dissimilarity analysis suggests that shape type is the dominant representation in all arrays

The average response across individual channels can exhibit weak category selectivity, but the categorical structure of the stimulus set may also appear in the pattern of activity distributed across the entire neuron population[20]. Therefore, we investigated how information about shape type and category was represented in the multichannel activity patterns. For each stimulus, we extracted averaged net activity (MUA level) and normalized high gamma power (LFP level) across time after stimulus onset for visually responsive channels. The dataset was randomly split into non-overlapping subsets (A and B) through 100 iterations for addressing variability. The multichannel activity patterns

of stimuli in set A were correlated with those in set B, and the resulting coefficients were averaged across iterations, yielding a $N \times N$ ($N = 54$, the number of individual stimuli) correlation matrix for each micro-array. These matrices were then converted into dissimilarity matrices (1-correlation). The resulting dissimilarity matrices (1-correlation, Fig. 5A) were correlated with behavioral dissimilarity matrices for the shape type and category dimensions as well as with the physical dissimilarity matrix based on the silhouettes (Fig. 5B) by means of Representational Similarity Analysis (RSA)[29]. For all microarrays, the multi-channel analysis revealed significant shape-based and silhouette representations in the MUA responses, but no significant correlation with the category matrix (Fig. 5C and Table 2, scatterplots of all behavioral and neuronal dissimilarities for each stimulus pair in Fig. S9). At the HG level, we observed similar results for arrays 3 and 4 (Fig. S11), but array 1 only correlated significantly with the silhouette dissimilarity matrix and array 2 only with the shape dissimilarity matrix (Table S3 and Fig. S11). Thus, the multichannel response pattern of all 4 arrays in LOC was predominantly shape-type. Moreover, the neural (MUA) dissimilarity matrices correlated significantly with both the perceptual and the physical dissimilarities. Interestingly, these population-level analyses suggest no contribution of category similarity, while the aforementioned single-channel analyses revealed many sites with an interaction between shape and category tuning. This dissimilarity analysis included all visually-responsive channels, but selecting only contacts with a significant main effect of shape type, category and/or a significant shape type x category interaction yielded highly similar results.

We quantified the shape type differences following ref. 30, who included perimeter and area as measures for "Aspect Ratio" to obtain a single dimension of shape. We then constructed the Aspect Ratio dissimilarity matrix by calculating the pairwise absolute differences in Aspect Ratio between all pairs of stimuli. The Aspect Ratio dissimilarity matrix correlated significantly with both the behavioral shape and the silhouette matrices (Supplementary Fig. S10). Moreover, the neural dissimilarity matrices of array 1 and 2 correlated strongly with the

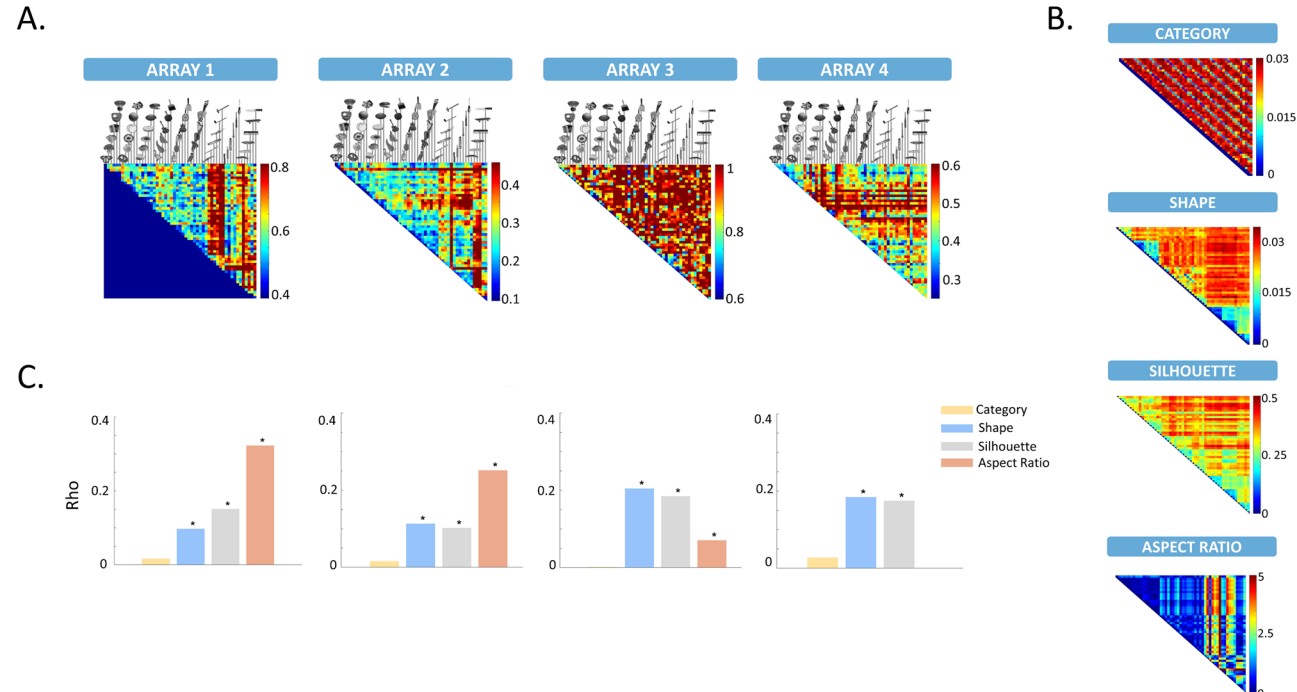

**Fig. 5 | Dissimilarity analysis for MUA. A** Neural dissimilarity matrices for all arrays based on the MUA responses. **B** Dissimilarity matrices for the shape and category dimensions as rated behaviorally, for the silhouette as calculated from the pixel-wise overlap between stimuli, and for the aspect-ratio. **C** Results of RSA for category-similarity (orange), shape-similarity (blue), silhouette-similarity (gray), and aspect-ratio (red). The asterisks indicate the significance of the correlation.

**Table 2 | Results of Representational Similarity Analysis (RSA) conducted on the MUA neural dissimilarity matrices**

| ARRAYS | Category | Shape | Silhouette |
|---|---|---|---|
| 1 | Rho = 0.02, *p* = 0.27 | Rho = 0.1, *p* = 0.00 | Rho = 0.15, *p* = 0.00 |
| 2 | Rho = 0.02, *p* = 0.27 | Rho = 0.11, *p* = 0.00 | Rho = 0.10, *p* = 0.00 |
| 3 | Rho = 0.002, *p* = 0.45 | Rho = 0.2, *p* = 0.00 | Rho = 0.18, *p* = 0.00 |
| 4 | Rho = 0.03, *p* = 0.16 | Rho = 0.18, *p* = 0.00 | Rho = 0.17, *p* = 0.00 |

The following key measures are reported: Rho (Pearson Correlation): Rho represents the Pearson correlation coefficient, following a permutation test ($n$ = 1000), quantifying the similarity between the neural dissimilarity matrices and the behavioral dissimilarity matrices; $p$: The $p$ value associated with the correlation coefficient, indicating the level of statistical significance.

Aspect Ratio matrix, while this correlation was weaker but still significant for array 3 (Fig. 5). In contrast, array 4 did not show any correlation between the neural and the Aspect Ratio dissimilarity ratio, suggesting that the neural responses recorded on array 4 may have been more strongly influenced by other shape features or even by non-shape stimulus properties (such as texture). This analysis therefore reveals more subtle differences in the shape representation between the four arrays which could not be captured with the single-channel analysis,

Next, we visualized the representation of the stimuli in the neural spaces of each array using MDS on the dissimilarity values. The 2D solutions of the MDS are shown in Fig. 6 (additional quantification of the MDS results is provided in Fig. S12). To evaluate the presence of clustering in each dimension, the stimuli were color-coded according to shape type (top row of Fig. 6) and semantic category (bottom row of Fig. 6). As an additional step to verify the existence of shape and/or category clusters within each array, we applied agglomerative hierarchical cluster analysis (Fig. S13). Shape clustering was evident with both methods in arrays 1, 2, and 4, with aspect ratio as an important factor mainly in arrays 1 and 2, while the MDS solution color-coded based on category did not exhibit a clear clustering. Array 3, on the other hand, did not exhibit strong clustering for the shape dimension, but when color-coded according to category, three exemplars of the category "animals" (rabbit, owl, and fish) were clearly separated from

the other stimuli (see Fig. S14 for the HG results, where a similar observation is made). The hierarchical cluster analysis corroborated this observation, since a subset of animal exemplars clustered together in the neural space of Array 3. Overall, these findings are consistent with the shape-based representations we found in the multivariate correlation analysis, but they also suggest the presence of some additional category information in array 3.

### Linear decoders detect reliably both category and shape information

The MDS analysis offers a representation of the stimuli in a limited number of dimensions in the neural space of the recorded population, but a decoder can utilize all the multidimensional information in a population. Moreover, decoding can be performed over time, which can also give insight into the temporal dynamics of the neural responses. Therefore, we trained linear Support Vector Machines on the neural responses per array in 100 ms bins (sliding window of 50 ms), and tested on each time bin of individual trials whether we could correctly classify either the shape type or the category. Figure 7A illustrates the temporal evolution of the decoding accuracy at the MUA level (as described in the Methods section) for the two decoders (shape type and category). In all 4 arrays, we could reliably decode shape type starting as early as 75 ms after stimulus onset for array 1, compared to 125 ms for array 2, 200 ms for array 3, and 175 ms after

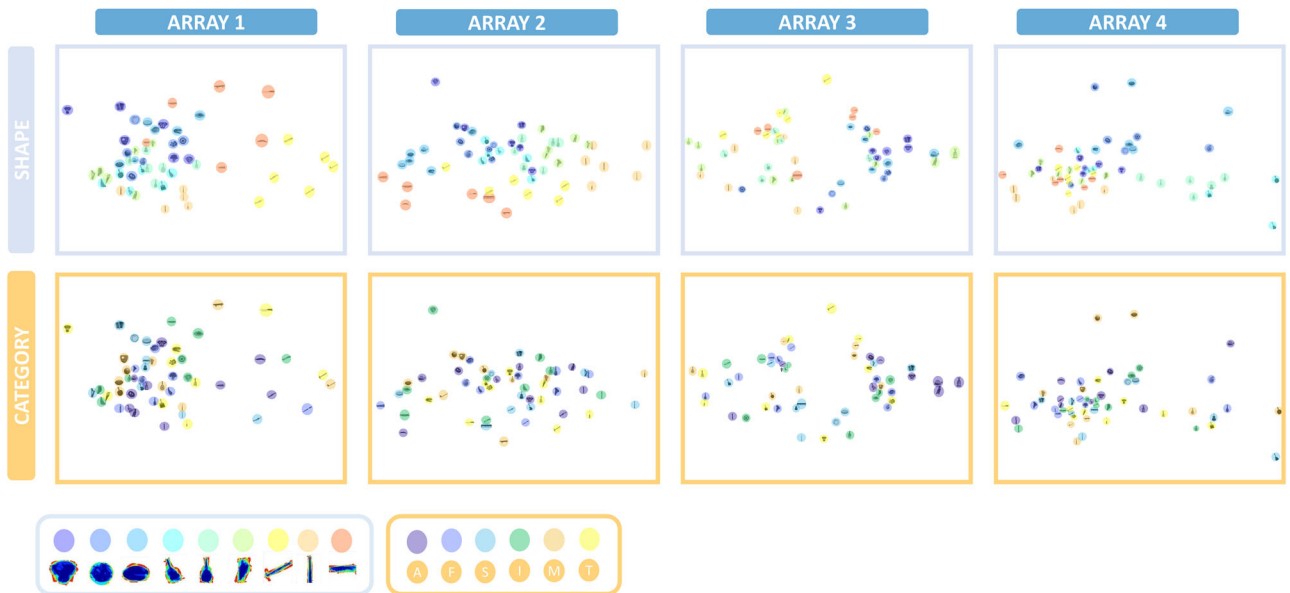

**Fig. 6 | Multidimensional scaling for the MUA neural dissimilarity matrices.** MDS performed on MUA neural dissimilarity matrices shows pairwise distances in a 2D space for each array. The 2D arrangements are color-coded first according to the 9 different shape types (upper panel), and then according to the 6 different semantic categories (lower panel).

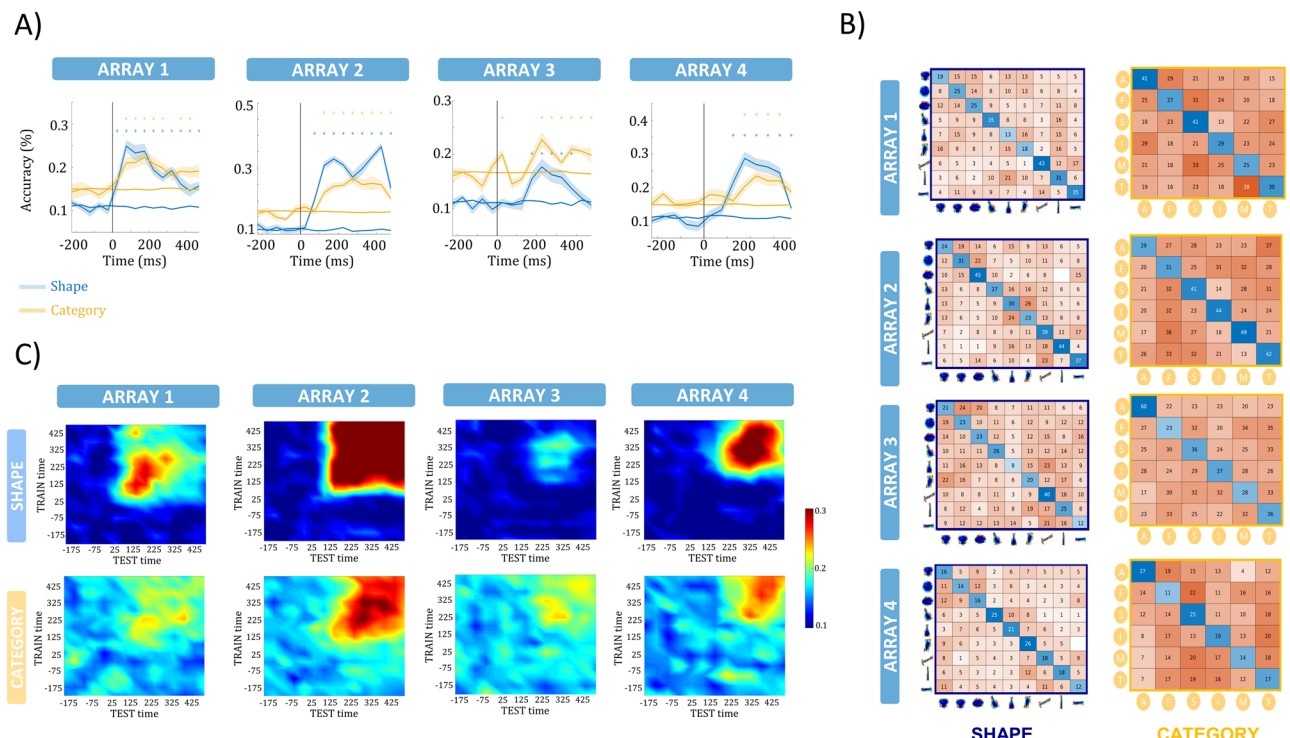

**Fig. 7 | Linear decoding of the MUA responses. A** Temporal evolution of the SVM decoding accuracy for the shape (blue) and the category (orange) dimension at the MUA level. The line represents the average across cross-validations, while the shaded region around the line indicates the standard error across the cross-validations. The asterisks indicate the significance of the accuracy. The horizontal orange and blue lines indicate the chance level for the category and shape dimensions, respectively, as computed from the permutation test of arbitrary groupings of the stimuli. **B** Confusion matrices are illustrating the performance of the decoding per class for the shape (upper panel) and the category (lower panel) dimension for a specific time-window (arrays 1,2: 75–275 ms, array 3: 175–275 ms, array 4: 125–225 ms) at the MUA level. The classification performance of array 3 for the category dimension is predominantly restricted to the "animals" category. **C** Generalization of the decoders over time for the shape (upper panel) and the category (lower panel) dimension. The y-axis corresponds to the TRAIN time window, the x-axis to the TEST time-window, and the colors to the accuracy level of the decoding.

stimulus onset for array 4 (Fig. 7A) (defined as the center bin within the first three consecutive bins achieving significant decoding accuracy). Furthermore, and in line with the previous analyses, array 3 also showed significant classification of category information, which was predominantly restricted to the "animals" category (see confusion matrix in Fig. 7B). Interestingly, however, despite the presence of primarily shape type representations on the other arrays, we also obtained significant classification of category on arrays 1, 2 and 4, which emerged almost simultaneously with the shape type classification. Thus, although neither individual channels nor the multichannel response pattern appeared to furnish any category information, a population of shape-selective neurons in human visual cortex contained reliable information about object category (Fig. S15 for HG decoding).

As a strong control to ascertain that shape information was not contaminated by category information and vice versa, we decoded shape type using different categories for training and testing, and we decoded category using different shape types for training and testing. Overall, the decoding of both shape type and category remained significant although category decoding was reduced in arrays 1 and 4 (Fig. S16). Therefore, the category decoding we observed was not exclusively driven by shape differences, even more, it was able to generalize across differences in shape. An additional control for the decoding result consisted of arbitrary groupings of the stimuli into 6 (as a control for the decoding of the 6 categories) and 9 groups (as a control for shape type decoding), on which we trained a decoder to classify between these groups. A permutation test with 100 repetitions showed chance performance in both cases (Fig. 7A), indicating that our decoding results were not the result of arbitrary shape differences.

To further investigate the predominant association of category information with the "animals" category, we conducted additional analyses by removing the "animals" category and performing the decoding again (Fig. S17). The decoding accuracy for arrays 1 and 2 at both the MUA and HG levels remained unaffected. However, a noticeable decline in both accuracy and significance was observed for array 3 at both the MUA and HG level. These findings were consistent with the observations from the confusion matrices (Figs. 7B, S15B), emphasizing that the category information was predominantly restricted to the "animals" category for array 3.

We assessed the generalization of the decoders over time (Fig. 7C). The shape and category decoders were trained using 100 ms time windows, and then tested on every 100 ms window that followed or preceded the training bin. Each window was then shifted by 50 ms. The decoding accuracy of array 2 generalized over the entire stimulus duration for both shape type and category, suggesting a very stationary population representation emerging early after stimulus onset, while arrays 1, 3, and 4 exhibited a more transient generalization of the classifier. At the HG frequency range (as depicted in Fig. S15), we observed, on average, highly similar decoding performance, albeit with lower levels of accuracy.

To relate our findings to previous studies using deep neural networks[25,31–33] to model responses of macaque ventral stream neurons, we conducted a forward pass through VGG-19 and ResNet-50 for each of the 54 stimuli, capturing the activation of weights in each layer. This process generated a matrix for each layer with dimensions equal to the nodes of the corresponding layer times the stimulus set (54). Subsequently, we computed the correlation between the activation patterns of the different stimuli, resulting in an Representational Dissimilarity Matrix (RDM) with dimensions $N \times N$, where $N$ represents the number of stimulus conditions. Using RSA, we quantitatively compared CNN representations per layer with both behavioral ratings and neural data. Figure 8 illustrates the correlation between CNN and behavioral RDMs (Fig. 8A), between CNNs and MUA RDMs (Fig. 8B), and between CNNs and HG RDMs (Fig. 8C) for the two architectures. The highest correlation between the CNNs and the neural data was found in the intermediate layers (12–15 for VGG-19 and 25–48 for ResNet-50). (See also Tables S4, S5).

## Discussion

We recorded selective MUA and LFP responses to images of objects on four microelectrode arrays in the human Lateral Occipital Complex. Both single-channel and multi-channel analyses revealed robust encoding of shape type and a very weak representation of category, consistent with previous electrophysiology studies in nonhuman primates. However, from each neuronal population, we could reliably classify category using linear decoders. Furthermore, single-channel analyses revealed that many channels showed interactions between the shape and category dimension, demonstrating the added value of single-channel information to reveal the tuning complexity underlying object processing in the human ventral visual stream.

While a large number of studies have been published on shape-sensitive cortex in humans using fMRI, electrophysiological data on the shape selectivity of human visual neurons remain scarce. Decramer et al.[27] showed for the first time single-unit and HG selectivity for images of objects and line drawings of objects (the LOC classic localizer) in lateral occipitotemporal cortex, including receptive field estimates (on average 22° diameter centered on the fovea) and selectivity for disparity-defined curved surfaces. A subsequent study[34] reported robust face-selective responses at short latencies, which also occurred for feature-scrambled and face-like stimuli. In the same study, a few channels also showed body selectivity in close proximity to the face-selective channels. Compared to these two previous studies, we recorded from considerably larger populations of neurons across a more extensive part of the LOC, with a stimulus set in which the dimensions of shape type and category were orthogonalized. Our data confirm and clarify the abundant visual selectivity in this region, since on average 62% of the channels were visually responsive, while 67% of those were significantly stimulus-selective. Note that the average 2D-shape selectivity index we found (0.72) was comparable to reported shape selectivity in macaque area TE (0.65)[35], and that size and position invariance of shape preference was furthermore present in other experiments using the same arrays (a finding which will be addressed in a separate publication). The high incidence of stimulus selectivity is striking given that the use of multielectrode arrays precluded optimizing the stimulus to each recording site (e.g., position, size) and that each array only sampled from a 4 by 4 mm area of cortex. On the other hand, chronic multielectrode recordings of MUA (i.e., large and small action potentials) may furnish a more unbiased sampling of neuronal activity in the recording area, which is crucial for relating our findings with invasive recordings to fMRI results.

We used the same stimulus set and analyses as in the event-related fMRI study of ref. 24, who reported a transition from shape to category-based representations along the posterior to anterior direction in the ventral visual stream. While the early visual areas provide a purely shape-based representation correlating with the physical similarities between the stimuli, and the higher-level areas (in prefrontal and parietal cortex) provide a more category-based representation, several intermediate regions in or near the LOC represented both shape- and category information. Here, we not only could confirm the fMRI results, but also clarify the underlying neuronal selectivity of these combined shape/category representations. We mainly observed significant interactions between shape type and category on individual channels of every array. These interactions occurred in two types. The first type of shape-category interactions were responses to a small number of exemplars of a single category, as in array 3. However, on the other arrays we found channels in which the shape type preference differed between the categories tested, most likely due to a selectivity for small shape or texture differences between the members of a given shape type. For example, the owl and the grapes of shape type 6 differ in shape and texture, and evoke marked response differences in the

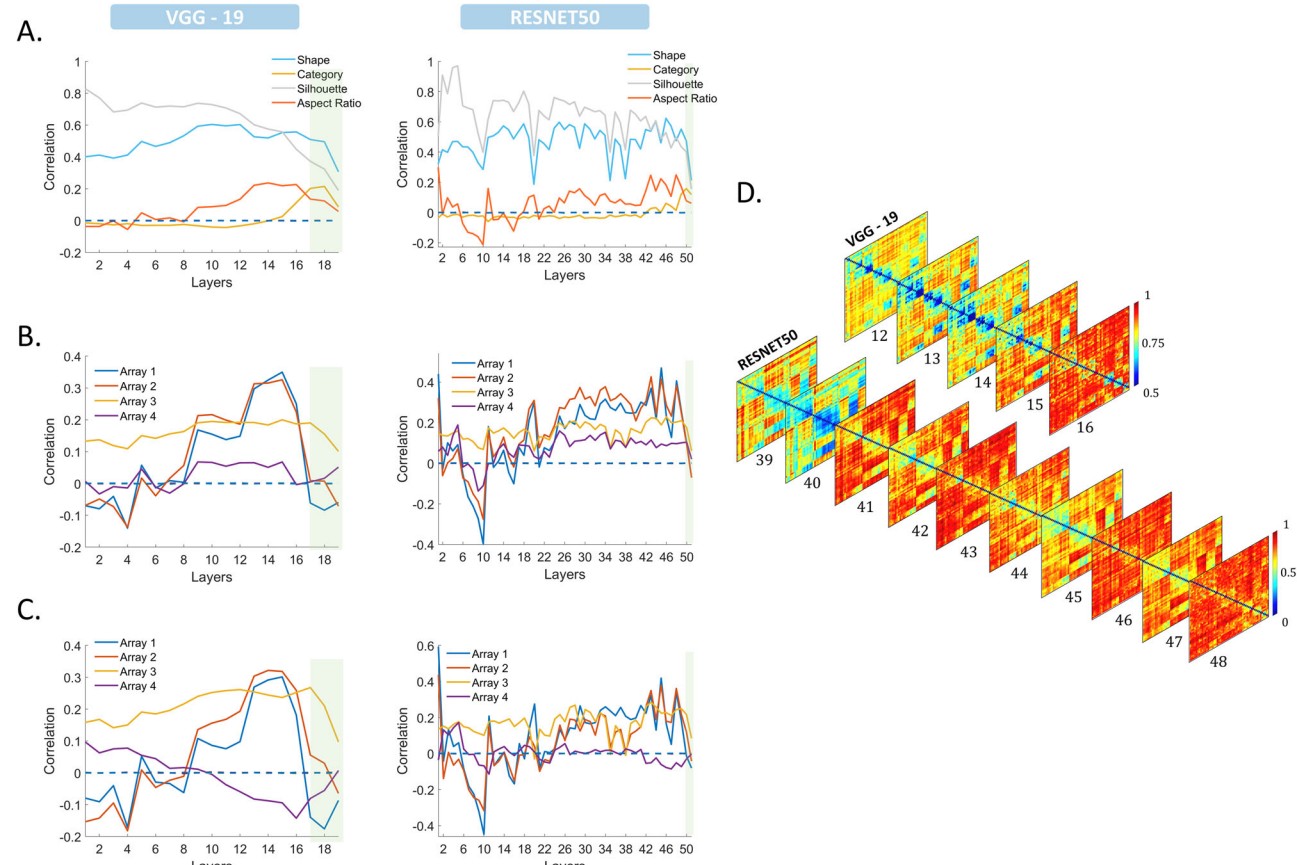

**Fig. 8 | CNNs. A** Correlation between CNN RDMs per layer (VGG-19 on the left, ResNet50 on the right) and behavioral RDMs for shape (blue), category (orange), silhouette (gray), and aspect ratio (red). The horizontal axis indicates network depth, and the vertical axis indicates correlation (Spearman's ρ). The green shading indicates the fully connected layers. **B** Correlation between CNN RDMs per layer and neural MUA RDMs for all arrays (Array 1: blue, Array 2: red, Array 3: orange, Array 4: purple). The green shading indicates the fully connected layers. **C** Correlation between CNN RDMs per layer and neural High-Gamma RDMs for all arrays (Array 1: blue, Array 2: red, Array 3: orange, Array 4: purple). The green shading indicates the fully connected layers. **D** Color plots of the RDMs for the intermediate layers of both VGG-19 and ResNet50 architectures, which correspond to the layers with the maximum correlation with the neural RDMs.

example illustrated in Fig. 2C. These interactions remain unnoticed in population-level analyses such as fMRI. Furthermore, the interactions were less prevalent with HG responses than with MUAs, suggesting that measurements of smaller populations of neurons are more likely to detect such interactions.

It is noteworthy that we measured robust stimulus selectivity in MUA and in HG responses, but not in the event-related potential (the ERP). Overall, the intracortically measured HG responses were highly similar to the MUA responses (consistent with ref. 27), which is an important observation for future invasive studies in humans. In our decoding results, MUA and HG behaved similarly when decoding shape type, but arrays 2 and 4 showed stronger decoding of category in HG compared to MUA. Future studies should investigate the significance of these HG-MUA dissociations in population-level analyses.

Array 3 demonstrated a clear preference for animal images compared to other objects. Considering this observation and its more dorsal positioning, it is highly likely that Array 3 was located within the region commonly referred to as LOTC-body in fMRI studies. The preference for animals on array 3 was the only category-like (i.e., responding to certain exemplars of one category) representation that was visible at the level of individual channels, whereas individual channels of all other arrays at most showed interactions of the category dimension with shape type. Intriguingly, even multi-channel analyses (dissimilarity analysis or hierarchical clustering) suggested that shape type was the dominant factor in every array. The lack of an explicit category representation (in arrays 1, 2, and 4) is entirely in line

with a previous single-cell study in the macaque inferotemporal cortex[19,36]. The latter observation is by no means trivial since the homology between shape-sensitive lateral occipital cortex in humans and (different parts of) the macaque inferotemporal cortex is currently unknown. Moreover, testing object category representation in macaques requires extensive training, whereas object categories in humans are known since childhood. Therefore, the theoretical possibility exists that the ref. 24 fMRI results showing shape-category interactions reflected the presence of more categorical object representations in human extrastriate visual cortex in comparison to macaques. Our intracortical recordings refute this hypothesis. In contrast, a linear SVM analysis could reliably extract category information from the population responses of every array. Conceptually, our decoding analysis was equivalent to Multivoxel Pattern Analysis[37,38], with a limited number of responsive channels (spaced 400 micron apart) being equivalent to the visually-active voxels in the fMRI. Thus, in the high-dimensional space of our LOC arrays (with up to 94 responsive channels), we could extract category information even when no individual channel appeared to code these categories. These results are again in line with previous findings in macaque monkeys, showing that category information can be reliably (and to a similar level as in prefrontal cortex) decoded from the activity of a population of ITC, prefrontal cortex, and hippocampus neurons, and simulated neural networks despite the lack of explicit category coding in individual neurons[39,40].

The SVM approach revealed category-level information which was not apparent using RSA, MDS, or hierarchical clustering for all arrays.

Specifically, the RSA analysis demonstrated that the neural representations in the Lateral Occipital cortex were primarily driven by shape and low-level pixel-wise similarities, indicating that the neural responses were more sensitive to the shape of the stimuli. This discrepancy between methods may be due to the fact that the SVM is more sensitive to subtle differences in patterns of neural activity than these other techniques, allowing it to decode information that is not detectable through measures of representational similarity. These observations match well with the findings from the single-channel analyses, since most channels showed interactions between the factors shape type and category. One such channel would not suffice to decode category, but multiple channels with different interactions would, in the same way as viewpoint-invariant recognition can be obtained by sampling multiple view-tuned neurons[41]. Likewise, the SVM might use a combination of channels that show interactions between shape and category to make a reliable distinction between categories. In contrast, RSA can reveal the structure of the neural representations of stimuli, which can provide insight into how the brain processes and categorizes different types of information. Note however that a single 4 by 4 mm array samples neural activity from a small cortical region (equivalent to 4 fMRI voxels in most fMRI studies), which may at best represent a single category (such as 'animals' in array 3). In contrast, RSA is typically performed on a very large number of voxels or on behavioral ratings, which encompass all categories in the stimulus set. The limited spatial sampling area of an array may explain why we did not observe a significant correlation with the category dissimilarity matrix in array 3.

Together, these findings highlight the complexity of neural mechanisms underlying object processing and the importance of using multiple techniques to uncover these representations. While the individual recording sites showed strong shape tuning and only very limited category selectivity, we found a large neuronal diversity and distinct interactions between shape and category at the single-channel level in human LOC, whereas the populations of neurons showed significant decodable category information. The broader relevance of this diversity in tuning was demonstrated by the ability of classifiers to decode not only shape but also category.

## Methods

Study protocol s53126 was approved by the local ethical committee (Ethische Commissie Onderzoek UZ/KU Leuven) and was conducted in compliance with the principles of the Declaration of Helsinki, the principles of good clinical practice, and in accordance with all applicable regulatory requirements. All human data were encrypted and stored at the University Hospitals Leuven.

Data were collected from three adult patients (aged 24–58 years old, including two females and one male) with intracranial depth electrodes as part of their presurgical evaluation for drug-resistant focal epilepsy. Patient 2 was diagnosed with Neurofibromatosis type 1, without any intracranial tumors. At the age of 34, she suffered from a left occipital intracranial hemorrhage due to venous sinus thrombosis. Ethical approval was obtained for microelectrode recordings with the Utah array in patients with epilepsy (study number s53126). Written informed consent in all patients was obtained before the start of the study.

### Patients

Three patients were implanted with microelectrode arrays (Utah array) for research purposes to study the microscale dynamics of the epileptic network in the presurgical evaluation ("Microscale Dynamics of Epileptic Networks: Insights from Multiunit Activity analysis in neurosurgical patients with refractory epilepsy", Bougou et al., EANS 2023, Barcelona). Utah arrays were located in the occipital cortex adjacent to the clinical macroelectrodes, analogous to previous studies using micro-electrode arrays[27,34,42–44]. (MNI coordinates of the arrays are

provided in Table 1). Patients 1 and 2 were implanted with one array, while in patient 3, two arrays were placed. Target locations of intracranial electrodes were determined by the epileptologist and based on electroclinical findings and non-invasive multimodal imaging.

The arrays were only implanted if a craniotomy was performed for the placement of subdural grids, therefore, the implantation of the arrays did not lead to additional incisions. The arrays were placed in close proximity to the subdural grids to study the microscale dynamics of the epileptic network. This was clearly discussed with all patients during the preoperative consultation (~1–2 months before surgery) and the day before surgery. Arrays were inserted in or near the presumed epileptogenic zone (based on preoperative multimodal imaging). Therefore, the brain tissue at the implantation site was a potential resection site prior to the recordings. After analysis of the intracranial EEG, it was deemed that the array was not inserted in the actual epilepticogenic zone (in patients 1 and 2 a remote focal onset zone was detected, and patient 3 had multifocal epilepsy). Importantly, none of the patients has experienced complications related to the micro-electrode array. After 14 days the arrays were removed together with the other clinical intracranial electrodes in a second surgery.

The first two subjects (patients 1 and 2) underwent an MRI after removal of the electrodes to investigate potential complications related to electrode implantation. The postoperative CT scan (with electrodes) was fused with the postoperative MRI (after removal of the electrodes) using the Brainlab© Elements software, to examine any structural alterations due to electrode insertion (Fig. S18). Based on a review of the different types of MR images, no structural alterations (gliosis, ischemia, hemosiderin) were seen at the implantation site. Furthermore, clinical neurologic examination withheld no functional deficits after electrode removal. In a previous study, we showed the safety of Utah array implantations and could even measure intact functional MRI activations after array removal, which demonstrates that the brain tissue was functional at the implantation site[27]. Based on these previous observations and on the reassuring anatomical MR imaging in the first two patients, we did not further systematically organize postoperative MRI imaging.

### Microelectrode recordings

We used 96-channel microelectrode arrays (4 × 4 mm; electrode spacing of 400 microns; Blackrock Microsystems, UT) in all patients. The arrays were inserted with a pneumatic inserter wand (Blackrock Neurotech). Dura was closed above the array and the bone flap was placed on top to keep the array in place. Reference wires were placed subdural, ground wires epidural. The signal was digitally amplified by a Cereplex M head stage (Blackrock Neurotech), and recorded with a 128-channel neural signal processor (NeuroPort system, Blackrock Neurotech, Salt Lake City, UT, USA). In each recording session, MUA from all 96 channels was sampled at 30 kHz, and high-pass filtered above 750 Hz. The detection trigger of the MUA was set at the edge of the noise band. The LFP signals were recorded continuously with a sampling frequency of 1000 Hz. All patients stayed at the hospital for 14 days after implantation, but the data reported here was acquired in 1 recording session per array (day 12 for array 1, day 2 for array 2, day 2 for array 3, and day 14 for array 4).

### Stimulus presentation

Experiments were performed in a dimmed hospital room. We presented stimuli on a 60 Hz DELL-P2418HZM LED monitor using custom-built software. The patients fixated a small red square (0.2 × 0.2°) appearing in the center of the display at a viewing distance of 60 cm (pixel size 0.026 deg). The left or right pupil position was continuously monitored using a dedicated eye tracker (Eyelink 1000 Plus, 1000 Hz) in head free mode. Breaking fixation from an electronically defined 3° by 3° fixation window resulted in trial abortion. The experiment was controlled using Presentation software (Neurobehavioral Systems,

Berkeley, CA, USA). For data synchronization, we attached a photo-diode to the left upper corner of the screen, detecting a white square that appeared simultaneously with the first frame of the stimulus; this "photocell" was invisible to the patients. Patients performed either a passive fixation task (patient 1) or a variant of the same passive fixation with a distractor (patient 2, patient 3) (in which the patients were asked to press a button with their right hand whenever a distractor (red or green cross) appeared at the fixation point, randomly in ~2% of the trials). After a fixation period of 300 ms, individual stimuli were presented for 800 ms (array 1) or 500 ms (arrays 2, 3, 4), followed by an interstimulus interval of 100 ms.

## Stimuli

We first screened for visual responsiveness in the MUA using images of objects and line drawings of objects (LOC classic stimulus set) presented at the center of the screen and at several positions in both hemifields. For each channel, we quantified the strength of the response at the different stimulus positions. This allowed us to determine the optimal position in the visual field per channel. To account for the variability in the receptive fields of individual channels, we presented the stimuli at the fixation point. Since we only analyzed visually-responsive recording sites, the receptive field of each responsive site included the fovea. Therefore, stimulus position was not optimized for each individual channel. This approach allowed us to capture a broader representation of the neural activity across the array.

**LOC localizer—Classic.** This stimulus set consisted of intact and scrambled grayscale images of objects and line drawings of objects[4,27] (Fig. 1). After a fixation period of 300 ms, each stimulus was presented for 800 ms, 500 ms, and 250 ms for arrays 1, 2, and 3 and 4, respectively, followed by an interstimulus interval of 100 ms for arrays 1 and 2 and 150 ms for arrays 3 and 4.

**LOC localizer—Naturalistic.** This stimulus set consisted of intact and scrambled colored and grayscale naturalistic images (Fig. 1), which were presented for 500 ms followed by an interstimulus interval of 100 ms.

**Shape-category stimuli.** A stimulus set of 54 images (approximate diameter of 8 visual degrees) in which shape and category were dissociated[24]. This stimulus set contained 6 object categories (minerals, animals, fruit/vegetables, musical instruments, sports articles, and tools) where each category included 9 grayscale images with unique shape properties (shape type). Therefore, the category and shape dimensions were orthogonal since every category contained one stimulus from each of the nine shapes and every shape contained one stimulus from each of the six categories. We quantified the shape similarity of all stimuli using the formula described in ref. 30 (see also Yargholi and Op De Beeck, 2023[45]). We applied this formula to calculate the "Aspect Ratio," which represents a single dimension of shape, particularly the distinction between "stubby" and "spike" shapes. We hypothesized that this Aspect Ratio would correlate with our behavioral shape ratings. Using this quantification, we constructed the "Aspect Ratio" dissimilarity matrix by calculating the pairwise absolute differences in Aspect Ratio between all pairs of stimuli, and calculated the correlation between the behavioral dissimilarity matrices and the Aspect Ratio dissimilarity matrix.

## Data preprocessing

We analyzed all data using custom-written MATLAB R2020b (Math-Works, Natick, MA, USA) scripts and the EEGLAB toolbox[46].

**MUA.** We calculated net average MUA responses (in 50 ms bins) by subtracting the baseline activity (−300 to 0 ms before stimulus onset) from the epoch (50−350 ms after stimulus onset) in each trial ($r_i$).

**LFP.** To remove line noise, data were filtered with a combined spectral and spatial filter[47] which can eliminate artifacts while minimizing the deleterious effects on non-artifact components. A zero-phase Finite Impulse Response (FIR) bandpass filter between 2 Hz and 300 Hz was then applied to the data. Trials of which the broadband activity deviated more than twice the standard deviation were discarded. The LFP power was analyzed in the HG band (60−120 Hz). For every trial, the time-frequency power spectrum was calculated using Morlet's wavelet analysis[48,49] with a resolution of 7 cycles. The first and last 100 ms of each trial were discarded to remove any filter artifacts. Power was normalized per trial by dividing the power per frequency by the power for this frequency averaged over time in the 300 ms baseline interval before stimulus onset.

## Latency

We calculated the response latency for each visually responsive site using a method where we examined the net spiking activity in 25 ms bins and compared them to the baseline activity. The first of the three consecutive bins with a significant difference from the baseline was considered the response latency. Because the arrays were not implanted in the same anatomical location, and because other factors such as anti – seizure medications may have influenced the responses, we chose an analysis window based on the mean response latency across channels of each array. Specifically, we computed the mean latency across channels for each array (array 1: 65 ms, array 2: 123 ms, array 3: 186 ms, array 4: 113 ms). Subsequently, we identified the nearest 50 ms bin to this value and selected a window encompassing 50 ms before and 150 ms after for the remainder of our analysis. The final analysis windows were determined as follows: array 1 (25−225 ms), array 2 (75−275 ms), array 3 (125−325 ms), and array 4 (75−275 ms).

## Visually responsive sites

We acquired at least 10 correct trials per stimulus (ranging from 10 to 19 trials). To detect visually responsive MUA channels in the shape-category test, we compared the average activity across time during the baseline period (−300 to 0 ms before stimulus onset) with the average activity in a 200 ms interval after stimulus onset using a 1-way ANOVA. Channels with a significant increase in activity ($p$ value lower than 0.05 divided by the number of channels to correct for multiple comparisons) were considered visually responsive. For the HG responses, due to lower Signal to Noise Ratio, we performed the 1-way Anova between the baseline and the post-stimulus interval only for the two most preferred conditions per channel. We determined the preferred condition for each channel, by averaging the post-stimulus per condition, sorting them in a descending order, and selecting the first two conditions with the strongest responses.

## MUA normalization for LOC localizer

For comparison with ref. 27, the MUA responses to the LOC localizer stimuli were normalized according to their peak values. More specifically we first averaged the net responses across "intact" stimulus trials and found the peak value per channel. Then, the responses per channel for both "intact" and "scrambled" stimuli were divided by the corresponding peak value.

## Z-score normalization for shape-category stimuli

To visualize the MUA and HG responses, we employed z-score normalization by averaging the MUA activity across the post-stimulus interval and across trials, i.e., for each channel and for each stimulus separately. Subsequently, we performed a per-channel normalization of these averaged responses such that the mean and standard deviation across the 54 different stimuli was 0 and 1, respectively. The MUA and HG normalized responses were plotted (color-coded according to the z-score) following first the order of the mean responses for the shapes and then for the categories (orange square).

## Statistics

To assess the MUA and HG selectivity for intact vs scrambled images in the LOC localizer stimuli for each array, we calculated one-way ANOVAs on the normalized MUA responses across all visually-responsive channels of each array. For the shape-category test, a 2-way ANOVA with factors category and shape was performed per channel. For all factors that reached significance, we used Tukey's test with 95% confidence interval to correct for multiple comparisons. To evaluate the size of the effects we calculated the eta$^2$.

## Selectivity-index

We calculated the selectivity index (Swidth) to evaluate how strongly each channel responds to a preferred stimulus compared to non-preferred stimuli. This measure provides a quantitative measure of the degree to which a channel is tuned to a specific stimulus. It is defined as: $(n - \sum r_i / \max)/(n - 1)$, where $n$ is the number of individual stimuli (54), $r_i$ is the mean net response of one channel to stimulus $i$, and max is the largest mean net response[35,50].

## D-prime

Discriminability Index (d-prime) was calculated to quantify the effectiveness of each channel in distinguishing the target stimulus from non-preferred stimuli. It was calculated as follows[27]:

$$d' = \frac{\left(\mu_{pref} - \mu_{Non_{pref}}\right)}{\sigma} \quad (1)$$

Where $\mu_{pref}$ and $\mu_{Non_{pref}}$ denote the mean response to preferred and nonpreferred condition, respectively, and:

$$\sigma = \sqrt{(\sigma 2_{pref} - \sigma 2_{Non_{pref}})/2} \quad (2)$$

is the pooled variance of the two distributions. This measure explicitly takes into account the trial-by-trial variability of the response.

## Behavioral, physical similarity, and aspect ratio dissimilarity matrices

**Shape and category.** We used the similarity judgments for the shape and category dimensions rated by a group of participants in ref. 24 to construct shape and semantic category models by means of behavioral shape and category dissimilarity matrices. The notably low correlation between these two matrices (−0.1, as reported in ref. 24) further demonstrates the orthogonality between the shape and category dimensions.

**Silhouette.** Similar to ref. 24, and ref. 51 pixel-wise similarities among images were computed in order to construct the physical dissimilarity matrix and evaluate the image low-level shape properties/image silhouette. Specifically, for all pairs of stimuli we calculated the squared differences for each pixel, summed these squared differences across all pixels, then obtained the square root of this sum. The resulting value was normalized by dividing it by the square root of the total number of pixels. Lastly, we inverted the obtained measure to derive a pixel-based similarity index.

**Aspect ratio.** To objectively quantify differences in shape types, we utilized the formula developed by ref. 30, which captures the most important shape dimension structuring object space. This formula, based on the parameters of perimeter and area (as also outlined in Yargholi & Op De Beeck, 2023[45]), was employed to calculate the "Aspect Ratio." The Aspect Ratio serves as a single dimension of shape, emphasizing the distinction between "stubby" and "spike" shapes.

## Correlation multivariate analysis

A correlation multivariate analysis was used to analyze whether the multichannel activity pattern per array was category-based or shape-based[24,52]. For each visually responsive channel and each stimulus, the averaged net activity ($r_i$, at the MUA level) and the normalized high gamma power (at the LFP level) across time after stimulus onset were extracted. The full dataset was then randomly divided into two random and non-overlapping subsets of trials; A and B, which was repeated in 100 iterations to get a measure of variability. For each iteration, the multichannel activity pattern associated with each stimulus in set A was correlated with all the multichannel activity patterns of each stimulus in the set B. Then, the resulting correlation coefficients for each stimulus-pair were averaged across iterations, in order to extract a 54 × 54 correlation matrix for each microarray. Finally, the resulting neural matrices were converted into dissimilarity matrices (1-correlation) and were correlated with the behavioral dissimilarity matrices for the shape and category dimensions (Pearson r). As described in Op de ref. 51, permutation statistics were used to determine the significance of the entry-wise correlations between vectorized dissimilarity matrices across the corresponding entries of both vectors. Thus, we used a permutation test ($n = 1000$) to calculate the Spearman's correlation coefficient between the neural dissimilarity matrices and the behavioral dissimilarity matrices for shape and semantic category (RSA)[29]. For comparison, we also correlated the neural dissimilarity matrices with the physical dissimilarity matrices.

## Multidimensional scaling (MDS)

MDS was used to visualize the neural similarity structure per array by reducing the multi-channel activity patterns corresponding to each stimulus into a lower-dimensional space, while preserving similarities or distances between them. We used the Matlab function "mdscale" which performs nonmetric multidimensional scaling by transforming monotonically all the dissimilarities in the matrix and approximating corresponding Euclidean distances between the output points. We evaluated the goodness of fit for 1 until 10 dimensions by measuring the difference between the observed dissimilarity matrix and the estimated one (stress value). We used the 2-dimensional solution (even with poor goodness-of-fit) to visualize the level of similarity of individual stimuli. To quantify the relative distances within clusters in the 2-dimensional plots, we calculated the Euclidean distances between stimuli belonging to the same shape type and between stimuli belonging to the same category group within this 2D space. We also computed and compared the Euclidean distances within (intra-cluster) and between (inter-cluster) conditions for each array, separately for the shape and the category dimension. To assess whether these distances differed significantly between intra and inter-clusters, we conducted t-tests on the distances with factors "inter" vs "intra".

## Agglomerative hierarchical cluster analysis

We used agglomerative cluster analysis on the neural dissimilarity matrices, to identify whether the neural responses to different shapes and/or categories in each array cluster together in meaningful ways. This involved treating each observation as a separate cluster and iteratively merging clusters based on their similarity until the stopping criterion was met (maximum 10 clusters were allowed). The analysis was performed using the MATLAB function "linkage", with the nearest distance default method.

## Linear decoding

To further investigate the multichannel responses we applied a linear Support Vector Machine (SVM) to classify sample vectors of which the entries consist of the per-channel net activity (at the MUA level) or the gamma power (at the LFP level) averaged over a time window of

100 ms. We focused on visually responsive channels (net multiunit activity (MUA) and normalized high gamma). To explore the dynamics of decoding accuracy, we applied a sliding window approach with a 100 ms duration, shifting it in 50 ms steps across the trial duration. Before training and testing the decoder, we performed z-score normalization on the data. The multiclass decoder was trained separately for each time-window, to find the hyperplane that separates the data from either the 9 individual shapes, or the 6 individual semantic categories. To prevent data leakage across trials, a cross-validation scheme was employed, dividing the dataset into 10 folds[53]. The training and testing phases were strictly independent, ensuring that the model's performance was evaluated on unseen data. Class labels of testing trials were excluded during training to ensure unbiased prediction. To assess the significance of the decoding accuracy, a paired t-test was performed, comparing the observed accuracy against the null hypothesis of random chance. We considered a decoding accuracy as significant if it exceeded the threshold of $p < 0.05$. To evaluate whether the SVM decoder generalized over time, we first allocate entire trials to the train and test set, we trained a decoder for each window shift and then tested on the activity across all other time windows for the duration of the whole trial. We implemented additional controls to prevent any leakage of category and shape information during training and testing. Specifically, for shape decoding, we used different categories for training and testing, and likewise, for category decoding, we used different shapes for training and testing to minimize the risk of contamination. This approach prevented the potential influence of shape-related information on category decoding and vice versa. As a last control measure, we implemented arbitrary grouping of stimuli and conducted a permutation test. This involved randomly organizing our stimuli into 6 groups to emulate a control for the 6 categories, each containing 9 exemplars. Subsequently, we trained a decoder to classify between these groups. This process was iterated 100 times. Additionally, we repeated the permutation test with stimuli arbitrarily grouped into 9 sets, each comprising 6 exemplars, as a control for the 9 distinct shape types.

### Convolutional neural networks (CNNs)

We utilized pre-trained CNNs, specifically VGG-19 and ResNet50, trained on the ImageNet dataset. Each CNN architecture comprises multiple convolutional layers, followed by pooling operations and fully-connected layers. We performed a forward pass for all stimuli through both networks, extracting activation weights for each layer. This resulted in a matrix with dimensions nodes per layer × number of stimuli. Subsequently, we used these weights to calculate representational dissimilarity matrices (1-correlation) for all layers. To generate these dissimilarity matrices, we correlated the activations associated with each stimulus across all pairs of stimuli. This process yielded an $N \times N$ correlation matrix, where N represents the number of stimuli (54). We then converted this correlation matrix into a dissimilarity matrix (1-correlation). This approach allowed us to determine the correlation (Pearson) between the CNN dissimilarity matrices for all layers and the neural dissimilarity matrices. We identified the layers where this correlation reached its maximum value. Additionally, we conducted similar analyses by comparing the CNN dissimilarity matrices per layer with the behavioral dissimilarity matrices for shape and category, as well as the silhouette and aspect ratio matrices.

**VGG-19.** VGG-19[54] achieved the highest ranking for single-object localization in ILSVRC 2014 and secured the second position in image classification[55]. This CNN architecture is composed of 19 weighted layers, including an extra softmax read-out layer for classification. Specifically, it consists of 16 convolutional layers separated by five max-pooling layers, with the final three layers being fully connected.

### ResNet50

ResNets are a group of deep architectures that won the ILSVRC classification task in 2015[56]. ResNet50 uses a two-branch structure and consists of 50 stacked "residual units." These units employ a split-transform-merge strategy, performing identity mappings alongside $3 \times 3$ convolutions and rectification. A key feature of ResNet50 is its use of identity mappings, forcing the preservation of features instead of learning new representations at each layer. In the final steps, three layers handle average pooling, transition to 1000 dimensions with fill connections, and conclude with softmax classification.

## Data availability

The datasets generated during and/or analyzed during the current study are available from the corresponding author on request. Source data are provided with this paper.

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

## Acknowledgements

We are indebted to all patients who participated in this study. We thank Stijn Verstraeten, and Anaïs Van Hoylandt for technical assistance. We would like to thank Dr. Elahe' Yargholi for computing the Aspect Ratio measure for our stimuli and Dr. Steven Smeijers for the evaluation of the postoperative MRI and CT scans. This work was supported by Fonds Wetenschappelijk Onderzoek (FWO) grant G.0B6422N, KU Leuven grants C14/18/100 and C14/22/134, and HBP SGA3 945539. M.V. is supported by FWO 1169321N. T.T. is supported by FWO (senior clinical researcher; FWO 1830717N).

## Author contributions

T.T., P.J., and H.O.D.B. conceived and designed the experiment. T.T. and M.V. planned and performed arrays placement surgery. M.V. performed the recordings and was responsible for all clinical trial related activities.

V.B. performed the data analysis and wrote the manuscript. T.T. and P.J. supervised and guided the study. W.V.P. selected the patients and performed the presurgical planning of placement of electrodes. All authors reviewed and edited the manuscript.

## Competing interests

The authors declare no competing interests.
