## [Peer Review File · Nature Communications]

Neuronal tuning and population representations of shape and category in human visual cortexREVIEWER COMMENTS

Reviewer #1 (Remarks to the Author):

The study by Bougou et al- although based on a small number of patients- provides an informative window into human object recognition in its use of multi-electrode array recordings in human object areas. Thus, I find the paper to provide a significant advance in our understanding of the neuronal mechanisms of human object recognition. However, especially because of the power of their recordings- the analysis of the data is to be too limited. Furthermore, the conclusions, specially concerning categorical information, seem overstated at this point- requiring further analysis to strengthen them. I list my concerns in detail below.

Major points

1. The authors attribute the lack of object vs scramble selectivity in some arrays to the limited stimulus set. However, it is more likely that this is due to the anatomical location of some arrays relative to object areas. It is a pity the authors didn't conduct a functional LOC of individual patients using fMRI – which could easily resolve this problem. One hint that indeed anatomical location (i.e., different levels in the cortical hierarchy) is the source of the problem is the finding that the latency of responses differed across arrays. Previous studies show that LOC responses typically lag behind early cortical representations. It will be interesting to correlate the latency of response of the neurons in each recording channel with their object vs scramble selectivity index.
2. As the authors note a well-established categorization in high order visual cortex is based on face and body categorization. This effect may explain the observed animal-preference. It seems that in their data set some animal images contained faces while others have not. It will be interesting to examine whether the animal selectivity subdivided along the face vs body subdivision in this data.
3. The data set offers a rare opportunity to compare three measures of cortical activity- Multi unit, High frequency power and LFP-obtained from the same recording channel (see also my comment below). It will be interesting to compare the tuning curves of the different contacts across these three signals and compare their similarity and differences- e.g., in their tuning selectivity.
4. The correlation of the RDMs for behavioral vs neuronal responses is elegant but shows a rather low correlation values. Unfortunately, the entire result is compressed into three averages. It is important that the authors show the findings in more detail. For example- a scatter plot showing the relationship between the neuronal dissimilarity vs the behavioral dissimilarity measures for each pair of shapes will be helpful. Also, it will be important to compare relevant contacts- i.e., contacts that actually show a general signature of object vs scramble selectivity with contacts that are less selective. Again- a scatter plot here could be informative – i.e., plotting for each pair of images- the similarity between their neuronal and behavioral distance on one axis and their average object vs scramble selectivity on the other.
5. In the decoding results, I am worried that some of the categorization information may be contaminated by shape similarity. It is important to make sure that category and shape information are truly orthogonal in the analysis. For example- examine the category decoding on a more restricted set of stimuli that, on the one hand were judged as belonging to the same category, but, critically, were judged to be highly dissimilar shape wise. Similarly, the authors should test the effect of excluding all face and body containing images from their decoding analysis.
6. Important information available to the authors but for some reason neglected is to examine the relationship between anatomical and functional distances of pairs of channel recordings. Could it be that neighboring contacts show more similar shape tuning? The authors should analyze this issue e.g. through a scatter plot in which the anatomical distance between contact pairs should be plotted against their tuning dissimilarity.
7. The authors claim that the semantic categorization and shape selectivity were orthogonalized- however it is not clear how they quantitatively verified this and to what extent they ruled out possible correlations between shape and category.
8. The authors equate shape and physical similarities however these could easily be dissociated through e.g., size or location changes of the shapes. If the authors have relevant data in which identical shapes were presented in such physically dissimilar manner, they should report these results. If not- they should clarify this point more explicitly.
9. It seems that the authors term the high frequency power as LFP. This is confusing- usually the LFP term is reserved to the evoked mass potential- while the authors seem to call the change in

amplitude (power) of high frequency fluctuations the LFP. The terminology should be clarified. Furthermore, it will be interesting to examine the correlation between the "classic" LFP with the high frequency power measures.

10. The term the authors use in the abstract "spiking activity" is too ambiguous- since it applies both to single unit recordings as well as multi-unit activity. This is not merely semantics- since the pooling of multiple units with diverse functional properties in the multi-unit recordings may underlie, at least partially, the complex tuning properties the authors find. The authors should clarify and acknowledge this possibility in the discussion.

Reviewer #2 (Remarks to the Author):

Review of

Neuronal tuning and population representations of shape and category in human visual cortex

Neuronal response properties of cells in the human LOC are thoroughly analyzed and reported using a stimulus set based on shape and image category. The authors conclude that single site/electrode specificity was mainly centered on shape, and that category information could be decoded from population responses. I had some doubts about the last claim as will be specified below. nevertheless, the data are unique and the study provides interesting insight into the neuronal basis of human shape and category perception.

The findings indicate that neurons at single LOC sites that primarily carry shape information, but I am not yet convinced about the claim that the population of such neurons indeed specifically carries category information (see below). Furthermore, the link with previous animal studies and human neuroimaging should be developed more to clarify how this new work relates the two.

Main concerns

1) The study is introduced as a bridge between the known response properties of neuron in homologous areas of the monkey brain and fMRI response properties in humans. This comparison is underdeveloped. It does come back in the discussion (in rather general terms) but given its prominence in the motivation for the study I would have expected more specific hypotheses based on either human fMRI work or monkey electrophysiology, and a more systematic comparison of the current results with previous studies. Strengthening this link would also make it easier to see the paper's relevance and would make it feel a little less exploratory.

2) Related to this, the authors should relate their work to the studies by the lab of DiCarlo/ van Gerven (and others), who utilized artificial neural networks (ANNs) to model the neuronal responses in the temporal stream. Have the authors tried this approach? These previous studies could link the activity in areas along the temporal stream to specific layers in these ANNs, and I am curious how the results (i.e. ANN layers) of the present study would compare to these previous studies.

3) The scrambling method is a bit old-fashioned, there are now better ways to create control stimuli. E.g. scrambling introduces strong boundaries to that are absent from the non-scrambled stimuli. E.g. metamers can preserve e.g. 2nd order statistics. Please give a rationale for using this somewhat outdated method.

4) I am not convinced that the analysis using decoders of Fig. 7A provide evidence for shape and category selectivity. The authors showed 54 stimuli in total and there are a total of 237 multi-unit sites, i.e. the SVM lives in a very high dimensional space and one need to demonstrate that equivalent separations for arbitrary groupings of the same number of stimuli give rise to lower decoding accuracy. To give an example of this approach: if shape decoding reaches certain accuracy with all shape categories having 6 exemplars (as is the case here), one would like to see that a control analysis with 9 arbitrary groups of 6 stimuli that come from different shape categories (as a control) are decoded at a lower accuracy.

- The same approach should be used to support the claim of category tuning in array 3. Here the control analysis should choose arbitrary groups of 9 stimuli chosen from different stimulus categories and if there is category encoding, the prediction is that this scrambling method will give rise to poorer decoding.
- The stimuli do not only differ in shape and category but also in color in texture. Can the authors exclude that these features played a role in the tuning?
- What does it mean to "subtract chance level"? How was this done? Would it not be better to not subtract and show chance level in the figure?

5) The start of the results misses important information about the experiment. The reader would like to know why these electrodes were implanted in this region, why it is ethical to do so, e.g. were the arrays placed in a brain region that was later going to be resected?

- The reader would also like to know that the patients were awake (i.e. not undergoing surgery), how long these arrays stayed in place, and whether they were taken out in a second surgery. Some of this information is in Methods, but it is important to present these crucial aspects of the design before diving into the results.
- A related point is that the stimulus set should be explained in the beginning of the results. My impression is that for every category there was one shape per shape group. Hence, one exemplar for every category-shape combination.
- The authors used shape/ silhouette dissimilarity measures but this is not properly explained. Please consider readers that who not read these previous papers.
- Line 237 "In all 4 arrays, we could reliably decode shape type starting as early as 75 ms after stimulus onset for array 1, compared to 100 ms for array 2, and 200 ms after stimulus onset for arrays 3 and 4". Is that estimated by eye? Please use a method to determine latencies that is published and well controlled.

6) Line 122: the authors write "Had the arrays been presented with optimal intact and scrambled stimuli tailored to their specific preferences, the differences in selectivity among the arrays may have been more pronounced." This claim could be easily checked by removing one, two or more shape categories and thereby examine how the number of shapes in the set influences these measures.

7) Line 156 What is s_width ? Please explain the logic of measures that you use in the results section

8) Line 177. Does it make statistical sense to compare the number of units with an interaction to the number of units with a main effect?

9) The analysis related to figure 6 (MDS) is unsatisfactory and lacks statistics. Is the reader supposed to see clustering here and differences in the degree of clustering between the conditions? It would be important to support the qualitative differences between the upper and lower panels with an rigorous analysis.

10) The authors choose different analysis windows for the different arrays. The choice should be supported by a criterion. Just choosing something without explaining why makes it difficult to reproduce these findings by other researchers, and gives the impression of p-hacking.

Minor points

11) Some of the details in the figures are rather small making it necessary to zoom in on specific panels quite far. Can this be improved to make the figures more legible?

12) I wonder why the neural traces are spaced so coarsely in 50ms time bins. Is the data too noisy to show finer details?

13) The authors use numbers to refer to shape categories (e.g. "shape type 5,6 and 8"), but these numbers are not presented in the figures, so that the reader has to count. Readability would improve if the authors add the numbers to the figure panels.

14) In Figure 2, are the MUA examples the same channels and the LFP examples or are they

different? In general, are the tuning properties the same or similar for MUA and LFP on the same site, or do the two signals differ in this aspect? A step in this direction is given in Fig 4B which correlates the magnitude of spiking and high gamma LFP responses, but does not dive into the tuning properties. This comparison should be discussed and/or developed more.

15) In Figure 3, are the MUA and LFP sites individually ordered by z-score or is the shown topography the same? It would be nice if the relation between MUA and LFP would be observable. The colored brackets are difficult to see unless zoomed in quite far.

16) The interactions between shape and category appear to be pronounced (large proportion of the responsive cells). This makes the interpretation a bit complicated.

17) Figure 4 lacks y-axis labels. Please add them.

18) The dissimilarity analysis should be unpacked. I found it difficult to follow and relate to the other analyses.

19) There seem to be some remarkable differences between the time-based decoding results for MUA and LFP that are now summarized as LFP giving lower accuracy. Actually, the very sustained decoding accuracy of array2-MUA seems more transient in the LFP, while for array1 the LFP seems more sustained than the MUA, and for array4 the high decoding in MUA seems almost absent in LFP. I think there's more happening here, than just weaker accuracy. The discussions of these differences should be expanded. (Fig 7 vs fig S6)

20) In the discussion, the authors mention the spatial scope of voxels, arrays and neurons in relation to what is known about shape/category coding and what is shown here. I think this is an important point that goes back to trying to bridge human fMRI and monkey ephys. Based on earlier studies, what would we expect in terms of single site signals and array populations?

21) The authors conclude that: Together, these findings highlight the complexity of neural mechanisms underlying object processing and the importance of using multiple techniques to uncover these representations. While the population as a whole showed strong shape tuning and only very limited category selectivity, we found a large neuronal diversity and distinct interactions between shape and category at the single-channel level in human LOC.

Important control analyses are missing that are needed before this claim can be accepted. I'm not sure if I completely agree with that and I think a scheme showing a proposed (hierarchical) shape/category coding would help clarify this point. "The population as a whole" here seems to refer to the collection of single sites or the average of those. The authors themselves show and argue that the population as a whole (in the sense of the 'the dynamic non-averaged population activity) does in fact have decodable category information.

22) Line 413: "The fixation point included the average receptive field of the MUA for each array." What does it mean that the fixation point includes a receptive field? Do you mean that the receptive field included the fixation point? Was that true for all receptive fields?

23) Line 456: "For the high - gamma responses, due to lower Signal to Noise Ratio, we performed the 1 - way Anova between the baseline and the post - stimulus interval only for the two most preferred conditions per channel." This is statistically incorrect. The ANOVA assumes independently generated data, it is not correct to first make a selection (e.g. max) on the test statistic and to then do an ANOVA.

24) Legend of Fig. 2. What is the meaning of "channel 1/2"? These channels are not visible in the figure itself.

25) Legend of Fig. 7B. Upper/lower panel should be changed into left/right panel.

Reviewer #3 (Remarks to the Author):

Bougou et al

Neuronal tuning and population representations of shape and category in human visual cortex

Bougou et al perform exciting recordings of neuronal spiking activity in the human temporal lobe in 3 patients with drug-resistant epilepsy. Remarkably, the patient and the ethical committee allowed the implantation of an invasive Utah array devices, presumably outside the epileptogenic zone (see Patients section), which is extremely rare in patients with epilepsy. In one patient, there were even two Utah arrays implanted! Through these recordings, the investigators provide a characterization of the tuning properties of neurons in the human occipital cortex, an area of which we know essentially nothing about at the neurophysiological level. The authors analyze the data in terms of shape and category tuning, a concept that I have many doubts about. But independently of the complex notion of "category", what is particularly exciting about this study is the possibility of describing the responses of neurons in occipital cortex to different stimuli.

(1) The implantation of Utah arrays in non-epileptogenic tissue in the human brain is fascinating from a scientific standpoint and opens up a lot of ethical questions. It would be interesting to document any information the authors may have about the degree of damage introduced by the implantation of such large electrode arrays. To my understanding, there is no clinical justification for the implantation of such arrays. Most Institutional Review Boards have approved the implantation of electrodes for clinical reasons in epilepsy patients but have traditionally frowned upon the implantation of Utah arrays, except in a few intraoperative studies or in cases where the Utah array is implanted in tissue to be resected (Utah arrays have also been implanted in healthy tissue in paraplegic patients for the purpose of building brain-machine interfaces). Given that the data have already been recorded, there is no point in dwelling too much on this. But as a precedent, any information provided by the authors on this would be quite useful for the scientific community on whether this type of studies should be allowed in the future.

(2) I struggle with the use of the terms "shape" and "category" in this study.

(2a) I do not understand the sense in which the authors argue that shape and category are "dissociated". In lines 423-430, the authors explain that every "shape" contains stimuli from each "category" and every "category" contains stimuli from each "shape" (Figure 1B). This seems to be a rather loose description of the concepts of shape and category. In ref. 21, the authors specifically designed synthetic stimuli where shape and category were independent. That is, there were stimuli pairs that had a short distance (at least in pixel space) in shape but belonged to different categories. And there were stimuli that were very far (at least in pixel space) and belonged to the same category. Perhaps the stimuli in Fig. 1b loosely "appear" to have similar shapes in some colloquial sense along a column, but this is not a rigorous definition of shape.

(2b) One way in which studies have tried to separate categories from shape is by asking whether preferred stimuli for neurons can be better described based on category tuning or shape tuning and the answer in the inferior temporal cortex invariably is shape tuning (see studies of Earl Miller in ref. 21), without any evidence of categorical tuning. A recent example is Bardon et al PNAS 2022. Given that the current occipital locations seem to be well before inferior temporal cortex in the visual hierarchy, it would be remarkable to find any categorical tuning and no evidence for categorical tuning has been reported in monkeys in occipital or temporal cortex.

(2c) Another way in which studies have tried to compare image features is by using neural network models and asking whether a model can explain neuronal responses (e.g. Yamins and DiCarlo PNAS 2014). Again, the conclusion from these studies is that activity in inferior temporal cortex is well explained by shape tuning without any categorical information.

I would suggest removing all the discussion of shape and categories in the current study.

(3)

3a. It would be very nice to show individual examples of raster plots to evaluate the quality of the data.

3b. What are the units for the activity fig. 1a? Spikes/second? If so, why are the firing rates so low? Recordings in the macaque occipital cortex tend to show that individual neurons can fire tens if not hundreds of spikes/sec.

3c. Is Fig. 1A showing the average of all the channels in the array? If so, why is this interesting? Perhaps the issue here is the average over neurons with very different stimulus properties. The notion of LOC and "localizer" derived from fMRI studies is not very useful or relevant with real neurophysiological data. This type of averaging seems to occlude the interesting responses in the data. In fact, even though Fig. 2 does not show units either, the numbers on the y-axis are much larger than those in Fig. 1, suggesting that the averaging is at least partly probably to blame for the low activity in Fig. 1.

(4) Continuing with the challenge of defining shape and category with uncontrolled stimuli, it is difficult to interpret the category and shape tuning results in Fig. 2. I would find it much more instructive to show the responses of individual channels (individual neurons if the authors can do spike sorting, depending on the quality of the data) to all the stimuli and report the degree of selectivity across all the stimuli without making any distinction along the arbitrary dimensions of category and shape. I would even use the term "shape selective" in this analysis to refer to neurons that show different responses among all 54 stimuli.

(5) It is hard to assess the quality of the MUA data because the authors do not show any waveforms of action potentials, which is customary to do in the field. It would also be useful to do spike sorting to separate the activity of individual neurons.

(6) Many investigators using Utah arrays report that it is hard to discern spiking activity immediately and they have to wait days/weeks/sometimes even months to get high spiking activity quality. Given the paucity of Utah array data from the human brain for ethical reasons, it would be useful if the authors could document the quality of the data and the evolution of quality over time during the period of 14 days that the patients stayed in the hospital. At a bare minimum, it would be useful to report when the recording in the paper took place and to show quantification of the actual waveforms.

(7) Given the paucity of data from human occipital cortex neurons and the exciting recording performed here, plus the possibility that it will be very hard to get data from these areas in the human brain, it would be amazing to document these responses in much more detail. What is the latency of the responses? What is the size and shape of their receptive fields? How come all/most of the receptive fields seem to be centered in the fovea (this seems counterintuitive based on what we know from similar areas in monkeys)? Do nearby channels (neurons) share similar tuning properties (their last figure hints at this)?

REVIEWER COMMENTS

Reviewer #1 (Remarks to the Author):

The study by Bougou et al- although based on a small number of patients- provides an informative window into human object recognition in its use of multi-electrode array recordings in human object areas. Thus, I find the paper to provide a significant advance in our understanding of the neuronal mechanisms of human object recognition. However, especially because of the power of their recordings- the analysis of the data is to be too limited. Furthermore, the conclusions, specially concerning categorical information, seem overstated at this point- requiring further analysis to strengthen them. I list my concerns in detail below.

Major points

1. The authors attribute the lack of object vs scramble selectivity in some arrays to the limited stimulus set. However, it is more likely that this is due to the anatomical location of some arrays relative to object areas. It is a pity the authors didn't conduct a functional LOC of individual patients using fMRI – which could easily resolve this problem. One hint that indeed anatomical location (i.e., different levels in the cortical hierarchy) is the source of the problem is the finding that the latency of responses differed across arrays. Previous studies show that LOC responses typically lag behind early cortical representations. It will be interesting to correlate the latency of response of the neurons in each recording channel with their object vs scramble selectivity index.

REPLY: We were unable to perform fMRI in all subjects as some patients opted not to undergo additional imaging. To assess the anatomical placement of the arrays, we correlated the stereotactic coordinates of the arrays with known LOC (lateral occipital cortex) locations. Specifically, we compared the array coordinates with the object parcels as outlined in Julian et al. 2012¹. Our analysis revealed that all four arrays were located within the object vs. scrambled functional areas, as defined by these parcels. The figure below depicts the left (yellow) and right (red) LOC according to Julian et al. 2012¹, with black circles indicating the location of each array. We have changed the first paragraph of results on p. 4 accordingly, and added the figure in supplementary material (Figure S1).

We calculated the response latency for each channel using a method where we examined the net spiking activity in 50 ms bins and compared them to the baseline activity. The first of the two consecutive bins with a significant difference from the baseline was considered the response latency. Furthermore, we calculated the selectivity index for the scrambled vs. non-scrambled conditions, both in the classic and color experiments. We then examined the correlation between selectivity index (s-width) and latency to investigate any potential relationship between these factors. The results indicate that the correlation between s-width and latency was extremely low, and in fact, it did not show any significant relationship between both variables. We have included these correlation

values in the table below for reference. In cases where there is no value, the number of visually responsive sites was too limited to calculate a correlation.

	UTAH 1	UTAH 2	UTAH 3	UTAH 4
CLASSIC	R = -0.24	R = 0.29	R = -0.37	-
COLOR	R = 0.29	R = 0.13	-	-

These findings suggest that, at least in the context of our study, variations in response latency do not appear to be a driving factor for the observed lack of object vs. scramble selectivity. We would like to point out that latencies are influenced by multiple factors, including number of trials (statistical power) and anti-seizure (ASM) and other medication. Since we could not control for these factors, we prefer to not emphasize these latency differences between patients. Furthermore, we could only obtain LOC fMRI data after explantation of the array in two patients. Because of signal loss in LOC in one patient (most likely due to the presence of chronic blood products e.g. hemosiderin), these localizers are difficult to interpret.

2. As the authors note a well-established categorization in high order visual cortex is based on face and body categorization. This effect may explain the observed animal-preference. It seems that in their data set some animal images contained faces while others have not. It will be interesting to examine whether the animal selectivity subdivided along the face vs body subdivision in this data.

REPLY: We conducted an analysis to assess responses to animals with faces compared to those without faces, along with responses to inanimate stimuli. We plotted the average net multi-unit activity (MUA) for these conditions for all arrays. For three arrays, we did not observe significant response differences between animals with faces and those without faces, as illustrated in the figure below. Notably, for arrays 3 and 4, we observed a significantly stronger response to faces, as compared to the other conditions, which was more pronounced for array 4. It is however important to clarify that the face-body selectivity of this particular array is part of an ongoing study and was not the primary focus of the current research. This information has been added on p6 and in Figure S5A.

To assess potential confounding effects of animacy on the face/non-face categorization, we plotted the average net multi-unit activity (MUA) across channels based on taxonomic levels. The stimulus set included animals from various taxonomic classes: insects (3 exemplars), reptiles (1 exemplar),

birds (2 exemplars), mammals (1 exemplar), and fish (2 exemplars). Notably, for arrays 3 and 4, which exhibited a significantly higher response to stimuli with faces, we observed that this distinction was actually confounded by the taxonomic hierarchy. Specifically, the response for mammals was significantly different from other groups (Figure S5B).

3. The data set offers a rare opportunity to compare three measures of cortical activity- Multi unit, High frequency power and LFP-obtained from the same recording channel (see also my comment below). It will be interesting to compare the tuning curves of the different contacts across these three signals and compare their similarity and differences- e.g., in their tuning selectivity.

REPLY: We integrated broadband LFP into our analysis and evaluated the selectivity of the evoked-related potential (ERP). We first plotted the average ERP across all channels and conditions, separately for different shapes and categories. This initial visualization allowed us to assess selectivity. To establish our analysis window, we determined the time point corresponding to the initial peak of the ERP after the baseline for every visually responsive site. Subsequently, we calculated the median of all these time points and selected a time window of 25 milliseconds before and 25 milliseconds after that median point, indicated by the green line in the plots.

We identified visually responsive sites by detecting significant differences from baseline. Using a 2-way ANOVA test (similar to the one applied to MUA and HG), we determined the percentages of channels with a main effect of shape only, a main effect of category only, and those with significant interactions. Our analysis revealed channels with significant LFP selectivities only in arrays 1 and 2. For both arrays, most channels showed pure shape selectivity. In array 1, we also found some channels with significant interactions. This suggests that, even at this level, interactions are not as pronounced compared to the MUA and HG, highlighting once again the tuning complexity that appears at a smaller spatial scale. This figure is added as Supplementary Figure 7.

Furthermore, we examined the tuning selectivity between different modalities. We sorted the MUA responses in descending order separately for shape and category and then plotted the average HG and ERP amplitude across channels following that order. The figure below illustrates the ranking and the table provides slopes of the linear fits and their confidence intervals (CIs). Notably, we observed consistent tuning between MUA and high gamma for at least one of the two rankings (shape and category) in all arrays, with similar slopes and small CIs. However, the ranking for broadband LFP clearly indicates different tuning at this level. This figure is added as Supplementary Figure 8 and this paragraph is added on p. 7.

	SLOPE MUA	95 % CI MUA	SLOPE HG	95 % CI HG	SLOPE LFP	95 % CI LFP
ARRAY 1 - CATEGORY	-0.22	[-0.32,-0.11]	-0.1	[-0.32,-0.10]	-0.03	[-0.22,0.35]
ARRAY 1 - SHAPE	-0.12	[-0.17,-0.05]	-0.1	[-0.17 -0.03]	-0.04	[-0.05 0.13]
ARRAY 2 - CATEGORY	-0.18	[-0.23, -0.12]	-0.14	[-0.35, 0.06]	-0.0004	[-0.33,0.33]
ARRAY 2 - SHAPE	-0.1	[-0.13, 0-0.07]	-0.09	[-0.15, -0.02]	-0.004	[-0.11 0.10]
ARRAY 3 - CATEGORY	-0.17	[-0.28,-0.07]	-0.15	[-0.33,0.03]	0.02	[-0.27, 0.32]
ARRAY3 - SHAPE	-0.12	[-0.14,-0.09]	-0.07	[-0.14,0.00]	-0.01	[-0.12, 0.09]
ARRAY 4 - CATEGORY	-0.18	[-0.26, -0.10]	-0.1	[-0.37, 0.18]	0.002	[-0.27, 0.27]
ARRAY 4 - SHAPE	-0.1	[-0.13, -0.06]	-0.01	[-0.12, 0.09]	0.06	[-0.04, 0.16]

4. The correlation of the RDMs for behavioral vs neuronal responses is elegant but shows a rather low correlation values. Unfortunately, the entire result is compressed into three averages. It is important that the authors show the findings in more detail. For example- a scatter plot showing the relationship between the neuronal dissimilarity vs the behavioral dissimilarity measures for each pair of shapes will be helpful. Also, it will be important to compare relevant contacts- i.e., contacts that actually show a general signature of object vs scramble selectivity with contacts that are less selective. Again- a scatter plot here could be informative – i.e., plotting for each pair of images- the similarity between their neuronal and behavioral distance on one axis and their average object vs scramble selectivity on the other.

REPLY: The figure below presents scatter plots between all behavioral and neural dissimilarity matrices, providing a more detailed visualization of our results (the “aspect ratio” is explained in our reply to comment 2a of reviewer 3). The figure has been added in the supplementary material (Figure S9).

One justification for the low correlation values between neural and behavioral dissimilarity matrices is the substantial variability observed among the neural dissimilarity matrices for different arrays. The figure below illustrates the scatter plots of the dissimilarity matrices for all pairs of arrays, with correlation values displayed in the accompanying bar plot (stars indicate significance). Notably, with the exception of the correlation between arrays 1 and 2, which equals 0.5, the remaining correlations are all below 0.2. Consequently, it is reasonable to expect that achieving high correlations between each of these matrices and behavioral RDMs may be challenging. This variability among arrays, even when located in similar brain areas, underscores the necessity for invasive mesoscale recordings and highlights the unique intricacies of neural responses in different arrays.

Unfortunately, we could not analyze contacts that consistently demonstrated a general object vs. scramble selectivity alongside less selective contacts. This was due to the experiments being conducted on different recording days, leading to unstable activity recorded per channel. However, following the reviewer's suggestion, we conducted our dissimilarity analysis again, this time focusing solely on relevant contacts, defined as those displaying a significant effect of shape, category, or significant interaction. The figure below displays color plots of the resulting neural dissimilarity matrices, with the accompanying bar plots illustrating the correlation between these matrices and the RDMs (the "aspect ratio" is explained in our reply to comment 2a of reviewer 3). The outcomes of this analysis align completely with the results obtained when considering all visually responsive sites. It appears that the inclusion of only relevant contacts, as opposed to all visually responsive sites, does not significantly affect the overall findings of this particular analysis. For reference, the table below provides information on the number of channels that exhibited visual responsiveness, as well as the count of relevant contacts involved in this refined analysis. We have added (in the results on p7) a sentence stating that including only the relevant contacts did not change the main results. Unfortunately, we could not plot for each pair of images the similarity between their neuronal and behavioral distance on one axis and their average object vs. scramble selectivity because the neuronal distances between pairs of shapes are computed on all responsive electrodes per array.

	ARRAY 1	ARRAY 2	ARRAY 3	ARRAY 4
VISUALLY RESPONSIVE	51	94	27	65
SELECTIVE	30	79	23	44

5. Point 1: In the decoding results, I am worried that some of the categorization information may be contaminated by shape similarity. It is important to make sure that category and shape information are truly orthogonal in the analysis. For example- examine the category decoding on a more restricted set of stimuli that, on the one hand were judged as belonging to the same category, but, critically, were judged to be highly dissimilar shape wise.

REPLY: Thank you for this suggestion. Although the nature of the stimulus set aims to dissociate shape from category, we acknowledge that full orthogonality can never be guaranteed in a finite set. To further investigate possible contamination, we revisited the decoding process and implemented additional controls to prevent any leakage of category and shape information during training and testing. For shape decoding, we used different categories for training and testing, and likewise, for

category decoding, we used different shapes for training and testing to minimize the risk of contamination. This approach prevents the potential influence of shape-related information on category decoding and vice versa (added in Methods). Upon implementing these controls, we found that shape decoding remained robust and unaffected by the additional measures for all arrays, confirming the reliability of our shape decoding results. However, it's noteworthy that for arrays 1 and 4, category decoding accuracy was substantially reduced after applying these controls but still achieved significance in at least one time bin. In contrast, for arrays 2 and 3, category decoding accuracy remained reliable even after implementing these controls. This observation suggests that category decoding is not exclusively driven by shape differences but rather underscores that the population activity genuinely encodes category information for these arrays. The result of this control is now added on p. 9, Figure S16) in the new version of the paper (a figure with the results is also added below for the reviewer's convenience).

Point 2: “Similarly, the authors should test the effect of excluding all face and body containing images from their decoding analysis.”

We have conducted this analysis (see Fig. S17 in the new version) and observed that we could still decode category for most arrays, with the exception of array 4, after removing all animal images. We also repeated the decoding analysis with the new controls to address shape/category contamination, as mentioned earlier. After excluding the animal images, we observed that only array 2 maintained reliable category decoding. This outcome was largely expected because from the confusion matrices we have observed that category decoding in array 3 was primarily driven by the category animals, and is not uniformly distributed across the other categories. Additionally, this is consistent with human fMRI studies, where it has been shown that animacy distinction is the primary dimension in category representation (Bracci, et al. 2016², Kriegeskorte et al. 2008³).

6. Important information available to the authors but for some reason neglected is to examine the relationship between anatomical and functional distances of pairs of channel recordings. Could it be that neighboring contacts show more similar shape tuning? The authors should analyze this issue e.g. through a scatter plot in which the anatomical distance between contact pairs should be plotted against their tuning dissimilarity.

REPLY: We have examined the relationship between anatomical distances between electrodes and shape preference. For every channel, we first ranked the average responses to the different shape types. We then plotted the average shape type responses on all other channels while maintaining the same ranking. For each ranking, we calculated the slopes and goodness of fit for the linear fit. The slopes of the linear fits indicate to what extent shape preference was preserved on the other channels (a similar ranking analysis was performed for size tolerance, see comment 8). The same procedure was done for the category responses. The scatter plots below depict the relationship between all slopes derived from the ranking and the Euclidean distances between channels, separately for shape and category. None of the correlations reached significance. We discuss these nonsignificant correlations in the revised manuscript on p. 5. Since this aspect was also addressed in a previous paper (Decramer et al., 2019⁴) we would suggest to not include these figures in the revision.

In the second set of figures, we present scatter plots showing the relationship between the goodness of fit for the linear fit of all rankings and the Euclidean distances between channels. The correlations are indicated in the title of each plot.

The figure below provides an array overview of the arrays where each channel is color-coded based on its selectivity as derived from the 2-way ANOVA (A). In this visualization, it is evident that there is no distinct clustering of neighboring contacts according to selectivity. Instead, the distribution appears to be more random. In panel B, an overview is provided of the channels demonstrating animal preference for array 3. Specifically, the color-coded channels represent those with a significant main effect of category, with the highest activity observed for the animals.

Notably, these results demonstrate that there is no clear relationship between shape and category tuning and anatomical distances. These findings align with prior research by Decramer et al. 2019⁴, which also showed that shape preference differed markedly between neighboring contacts and indicated that the shape preference of human LOC neurons is clustered on a submillimeter scale, similar to the monkey ITC.

7. The authors claim that the semantic categorization and shape selectivity were orthogonalized- however it is not clear how they quantitatively verified this and to what extent they ruled out possible correlations between shape and category.

REPLY: The two dimensions are intended to be orthogonal by the design of the stimulus set, but it is indeed possible that they would interact. This can be checked by analyzing the behavioral ratings. In Bracci et al. 2016², it was reported that the two behavioral measures of similarity, semantic category and shape, showed no correlation ($r = - 0.01$).

8. The authors equate shape and physical similarities however these could easily be dissociated through e.g., size or location changes of the shapes. If the authors have relevant data in which identical shapes were presented in such physically dissimilar manner, they should report these results. If not- they should clarify this point more explicitly.

REPLY: We have indeed conducted experiments to determine shape, size and cue invariance with a different stimulus set. However, tolerance to stimulus transformations is currently under investigation as a separate study, which we are actively working on.

In the figure below figure , we provide one illustrative example channel per array for an experiment in which we presented 14 different shapes in four different sizes (arrays 1, 2, and 4). For array 3, a similar experiment was conducted, but in this case, 20 different bodies were shown in three different sizes. The figure below presents the responses for these example channels.

In panel A, the average Multi-Unit Activity (MUA) responses are displayed for each shape-size combination. Each column corresponds to different shapes (or bodies for array 3), and the colors represent different sizes. In panel B, we present the ranked responses for different sizes, for which the responses were sorted in descending order for the best size (the one with the strongest response), and then the responses for the other sizes were plotted following that order. If the shape preference is preserved for different sizes, the slopes of the linear fits are significantly lower than zero.

The results show that, for the example channels of arrays 1, 2, and 4, robust size invariance is found. However, this is not as clear for array 3. Overall, in arrays 1, 2, and 4, most channels exhibited both size and location invariance. The table below shows the slope and confidence intervals obtained from conducting a linear fit on the rankings. We are currently conducting a more detailed analysis of these findings, which will be presented in a forthcoming paper.

Since we acknowledge that tolerance of stimulus selectivity represents an important aspect, we discuss this on p. 10.

A)

ARRAY 1	Size 4	Size 1	Size 2	Size 3
SLOPE	-0.08	-0.03	-0.04	-0.05
95 % CI	[-0.09 -0.07]	[-0.07 0.01]	[-0.07 -0.02]	[-0.09 -0.01]
ARRAY 2	Size 2	Size 1	Size 3	Size 4
SLOPE	-0.05	-0.03	-0.02	-0.01
95 % CI	[-0.07 -0.03]	[-0.06 -0.01]	[-0.05 0.01]	[-0.05 0.02]
ARRAY 3	Size 3	Size 1	Size 2	
SLOPE	-0.05	-0.007	-0.004	
95 % CI	[-0.05 -0.05]	[-0.03 0.02]	[-0.03 0.02]	
ARRAY 4	Size 1	Size 2	Size 3	Size 4
SLOPE	-0.07	-0.04	-0.04	-0.02
95 % CI	[-0.08 -0.06]	[-0.08 -0.001]	[-0.07 -0.001]	[-0.08 0.03]

9. It seems that the authors term the high frequency power as LFP. This is confusing- usually the LFP term is reserved to the evoked mass potential- while the authors seem to call the change in amplitude (power) of high frequency fluctuations the LFP. The terminology should be clarified. Furthermore, it will be interesting to examine the correlation between the “classic” LFP with the high frequency power measures.

REPLY: We have rephrased this throughout the manuscript and use the term high frequency gamma power. See also our response to comment 3.

10. The term the authors use in the abstract “spiking activity” is too ambiguous- since it applies both to single unit recordings as well as multi-unit activity. This is not merely semantics- since the pooling of multiple units with diverse functional properties in the multi-unit recordings may underlie, at least partially, the complex tuning properties the authors find. The authors should clarify and acknowledge this possibility in the discussion.

REPLY: We have clarified this in the manuscript (on page 5) and addressed this in the discussion. We also show example single-unit responses in Figure S2. See also our reply to comment 4 of reviewer 3.

Reviewer #2 (Remarks to the Author):

Review of

Neuronal tuning and population representations of shape and category in human visual cortex

Neuronal response properties of cells in the human LOC are thoroughly analyzed and reported using a stimulus set based on shape and image category. The authors conclude that single site/electrode specificity was mainly centered on shape, and that category information could be decoded from population responses. I had some doubts about the last claim as will be specified below. nevertheless, the data are unique and the study provides interesting insight into the neuronal basis of human shape and category perception.

The findings indicate that neurons at single LOC sites that primarily carry shape information, but I am not yet convinced about the claim that the population of such neurons indeed specifically carries category information (see below). Furthermore, the link with previous animal studies and human neuroimaging should be developed more to clarify how this new work relates the two.

Main concerns

1) The study is introduced as a bridge between the known response properties of neuron in homologous areas of the monkey brain and fMRI response properties in humans. This comparison is underdeveloped. It does come back in the discussion (in rather general terms) but given its prominence in the motivation for the study I would have expected more specific hypotheses based on either human fMRI work or monkey electrophysiology, and a more systematic comparison of the current results with previous studies. Strengthening this link would also make it easier to see the paper’s relevance and would make it feel a little less exploratory.

REPLY: We elaborate on this in further detail on page 11 and in the discussion.

2) Related to this, the authors should relate their work to the studies by the lab of DiCarlo/ van Gerven (and others), who utilized artificial neural networks (ANNs) to model the neuronal responses in the temporal stream. Have the authors tried this approach? These previous studies could link the activity in areas along the temporal stream to specific layers in these ANNs, and I am curious how the results (i.e. ANN layers) of the present study would compare to these previous studies.

REPLY: We appreciate the reviewer's suggestion. To explore this, we employed pre-trained CNNs, namely VGG-19 and ResNet50, trained on the ImageNet dataset. This is now addressed on p. 9 – 10 and the results are shown in Figure 8 of the new version of the paper.

VGG – 19	LAYER	CORRELATION
ARRAY 1 - MUA	15	0.35
ARRAY 2 - MUA	15	0.32
ARRAY 3 - MUA	15	0.2
ARRAY 4 - MUA	15	0.07
ARRAY 1 - LFP	15	0.30
ARRAY 2 - LFP	14	0.32
ARRAY 3 - LFP	17	0.27
ARRAY 4 - LFP	1	0.10

RESNET50	LAYER	CORRELATION
ARRAY 1 - MUA	44	0.47
ARRAY 2 - MUA	42	0.42
ARRAY 3 - MUA	45	0.23
ARRAY 4 - MUA	4	0.19
ARRAY 1 - LFP	44	0.42
ARRAY 2 - LFP	44	0.38
ARRAY 3 - LFP	42	0.28
ARRAY 4 - LFP	4	0.17

3) The scrambling method is a bit old-fashioned, there are now better ways to create control stimuli. E.g. scrambling introduces strong boundaries to that are absent from the non-scrambled stimuli. E.g. metamers can preserve e.g. 2nd order statistics. Please give a rationale for using this somewhat outdated method.

REPLY: We now mention in the text on p. 4 that we merely showed scrambled control stimuli to relate our findings to previous fMRI studies using the LOC localizer. We agree with the reviewer that scrambling is not a very good control, and that the results are not essential for our findings.

4) Point 1: I am not convinced that the analysis using decoders of Fig. 7A provide evidence for shape and category selectivity. The authors showed 54 stimuli in total and there are a total of 237 multi-unit sites, i.e. the SVM lives in a very high dimensional space and one need to demonstrate that equivalent separations for arbitrary groupings of the same number of stimuli give rise to lower decoding accuracy. To give an example of this approach: if shape decoding reaches certain accuracy

with all shape categories having 6 exemplars (as is the case here), one would like to see that a control analysis with 9 arbitrary groups of 6 stimuli that come from different shape categories (as a control) are decoded at a lower accuracy.

- The same approach should be used to support the claim of category tuning in array 3. Here the control analysis should choose arbitrary groups of 9 stimuli chosen from different stimulus categories and if there is category encoding, the prediction is that this scrambling method will give rise to poorer decoding.

REPLY: We thank the reviewer for this suggestion. First, there is an important clarification to make regarding the dimensionality of our decoding analysis. We performed the decoding separately for each array, and the number of features in each decoding analysis corresponded to the number of visually responsive sites in that specific array. Therefore, we did not use the entire set of 237 features for each decoding analysis. Additionally, our decoding did not focus on the 54 individual stimuli. Instead, we performed decoding to classify between the 6 categories or the 9 shape types to which these stimuli belonged.

To address the reviewer's concern and provide a suitable control, we performed arbitrary grouping of the stimuli and conducted a permutation test according to the reviewer's suggestion. In this test, we randomly grouped our stimuli into 6 groups (to simulate a control for the 6 categories with 9 exemplars per group) and trained a decoder to classify between these groups. We repeated this process with arbitrary grouping of the stimuli into 9 groups of 6 exemplars as a control for the 9 shape types. We performed this permutation test 100 times for both cases.

The results of the permutation test demonstrated that, in both scenarios (6 and 9 arbitrary groups), decoding accuracy was at chance level, as indicated by the grey line in the new decoding figure provided below. The results of this control are added in Results on p. 9.

-Point 2: The stimuli do not only differ in shape and category but also in color in texture. Can the authors exclude that these features played a role in the tuning?

REPLY: This is correct, and as mentioned on p. 10 in the discussion these features may have played a role in the tuning. We discuss this further on p. 5 and p. 8.

- What does it mean to “subtract chance level”? How was this done? Would it not be better to not subtract and show chance level in the figure?

REPLY: The new decoding results are without normalization and chance level is shown.

5) The start of the results misses important information about the experiment. The reader would like to know why these electrodes were implanted in this region, why it is ethical to do so, e.g. were the arrays placed in a brain region that was later going to be resected?

- The reader would also like to know that the patients were awake (i.e. not undergoing surgery), how long these arrays stayed in place, and whether they were taken out in a second surgery. Some of this information is in Methods, but it is important to present these crucial aspects of the design before diving into the results.

REPLY: The requested information has been added to the methods section. The arrays were only implanted if a craniotomy was performed for the placement of subdural grids, therefore, the implantation of the arrays did not lead to additional incisions. The arrays were removed in a second surgery after 14 days in the operating room, during the same surgery for subdural grids (and depth electrode) removal. The arrays were placed in close proximity to the subdural grids to study the microscale dynamics of the epileptic network. For now, no clinical decisions are based on the array recordings, however, if important information would be present this may impact future clinical decision making in epilepsy surgery. This is clearly discussed with all patients during the preoperative consultation (approximately 1-2 months before surgery) and the day before surgery. We only insert the array in or near the presumed epileptogenic zone (based on preoperative multimodal imaging). If the seizure onset zone (as determined by multimodal imaging) is concordant with an intracranial lesion, the lesion is most often resected without prior intracranial EEG recordings. Therefore, since intracranial EEG is only implanted in so-called difficult (e.g. MR-negative, multifocal, discordant findings) refractory epilepsy, there is a real risk of the array (but of course also the intracranial recording electrodes) being outside the epileptogenic zone. In all patients discussed in this paper, after analysis of the intracranial EEG, it was deemed that the array was not inserted in the actual epileptogenic zone (in patient 1 and 2, another focal onset zone was detected, in patient 3, the epilepsy was multifocal). Importantly, none of our patients (more than 10 array insertions) has experienced complications thus far related to the array (i.e. infection, hematoma). In this paper we do not discuss the patients where we performed implantations in more dorsal occipito-temporal, parietal or frontal cortex.

We now added this information now in the beginning of the Results section on p. 4.

- A related point is that the stimulus set should be explained in the beginning of the results. My impression is that for every category there was one shape per shape group. Hence, one exemplar for every category-shape combination.

REPLY: We have added this information in the beginning of the Results section on p. 4.

- The authors used shape/ silhouette dissimilarity measures but this is not properly explained. Please consider readers that who not read these previous papers.

REPLY: We have added this information in the Methods, on p. 17-18.

- Line 237 "In all 4 arrays, we could reliably decode shape type starting as early as 75 ms after stimulus onset for array 1, compared to 100 ms for array 2, and 200 ms after stimulus onset for arrays 3 and 4". Is that estimated by eye? Please use a method to determine latencies that is published and well controlled.

REPLY: We have revised the values, specifying them as the central bin within the initial three consecutive bins demonstrating significant decoding accuracy. Array 1: 75 ms, Array 2: 125 ms, Array 3: 225 ms, Array 4: 175 ms (p. 9).

6) Line 122: the authors write "Had the arrays been presented with optimal intact and scrambled stimuli tailored to their specific preferences, the differences in selectivity among the arrays may have been more pronounced." This claim could be easily checked by removing one, two or more shape categories and thereby examine how the number of shapes in the set influences these measures.

REPLY: We apologize for the confusion. This sentence merely wanted to convey the message that, because we recorded with multielectrode arrays we could not optimize the stimuli to the neuronal selectivity of each electrode (as is typically done in single- microelectrode recordings). This sentence has been removed and has been replaced by the following sentence: "Although the degree of selectivity for image scrambling and the response latency differed between the arrays, the significantly stronger responses to intact images of objects compared to scrambled ones demonstrate that all arrays were located in shape-sensitive cortex, in agreement with Decramer et al. 2019⁴.

7) Line 156 What is s_width? Please explain the logic of measures that you use in the results section'

REPLY: We discuss this in the Results section on p. 5.

8) Line 177. Does it make statistical sense to compare the number of units with an interaction to the number of units with a main effect?

REPLY: We did not aim to compare the number of channels with an interaction to the number of channels with a main effect, but rather wanted to describe the types of effects we observed in our population. See p. 6.

9) The analysis related to figure 6 (MDS) is unsatisfactory and lacks statistics. Is the reader supposed to see clustering here and differences in the degree of clustering between the conditions? It would be important to support the qualitative differences between the upper and lower panels with an rigorous analysis.

REPLY: It is important to clarify that the MDS analysis method is designed as a visualization tool, i.e., to visualize the distance between high-dimensional data points in a 2D plot. The quantitative analysis of these matrices was the comparison of the neural dissimilarity matrices with the behavioral dissimilarity matrices.

Nevertheless, although a lot of quantitative information has been discarded in MDS plots (due to the 2D projection), one could try to quantify the relative distances in 2D MDS plots. To this end, we calculated the Euclidean distances between stimuli belonging to the same shape type and between stimuli belonging to the same category group within this 2-dimensional space.

Our analysis revealed that, for all arrays, there are differences in the Euclidean distances, primarily for the shape dimension. Specifically, we found that for certain shape types, the average distance is notably small, indicating that stimuli with these specific shapes tend to cluster together.

Concerning the category dimension, we did not observe significant differences in the Euclidean distances for arrays 1, 2, and 4. In the case of array 3, we noticed that the distances within the animal condition were notably smaller compared to the distances within the other categories. This observation aligns with the clustering of animal stimuli. Still, it's essential to note that this difference was not extremely pronounced, which may explain why the comparison with the category behavioral matrix did not reach statistical significance for any of the arrays. This analysis has been added as Supplementary Figure 12 and is discussed on p. 8.

The figure below depicts boxplots illustrating the distances within (intra-cluster) and between (inter-cluster) conditions for each array, separately for the shape and category dimensions. To assess whether these distances differed significantly between intra- and inter-clusters, we conducted t-tests. The results revealed significant differences only for the shape dimension of arrays 1 and 2. However, for the category dimension, there was no significant difference for any of the arrays. This outcome aligns with our expectations, as indicated by the figure above, where only array 3 exhibits some clustering, specifically within the animals condition. However, this clustering was not sufficient to yield statistical significance

10) The authors choose different analysis windows for the different arrays. The choice should be supported by a criterion. Just choosing something with explaining why makes it difficult to reproduce these findings by other researchers, and gives the impression of p-hacking.

REPLY: The arrays were not implanted at the same anatomical location in the different patients, and other factors (such as medication) may have influenced the latencies. Therefore, we chose an analysis window which was determined by the response latency of each array. We have added this explanation for the different analysis windows on p. 16.

Minor points

11) Some of the details in the figures are rather small making it necessary to zoom in on specific panels quite far. Can this be improved to make the figures more legible?

REPLY: All figures have been re-adjusted, details and font size were increased.

12) I wonder why the neural traces are spaced so coarsely in 50ms time bins. Is the data too noisy to show finer details?

REPLY: We chose 50 ms bins for the neural traces because they are sufficient to capture the dynamics of the Multi-Unit Activity (MUA). This choice was also consistent with the approach used in the study by Decramer et al. 2019, which had similar data. However, we acknowledge that recordings in human patients typically have less trials per recording session compared to studies in NHPs, which explains why we could not use a smaller bin size.

However, in response to your query, we conducted a decoding analysis with 25 ms bins (using the control described in comment 5 of reviewer 1) to investigate whether finer temporal granularity would yield different results. The results of this analysis, as shown in the figure below, align with those obtained using 50 ms bins.

13) The authors use numbers to refer to shape categories (e.g. “shape type 5,6 and 8”), but these numbers are not presented in the figures, so that the reader has to count. Readability would improve if the authors add the numbers to the figure panels.

REPLY: The numbers have been added in Figures 2, and 4.

14) In Figure 2, are the MUA examples the same channels and the LFP examples or are they different? In general, are the tuning properties the same or similar for MUA and LFP on the same site, or do the two signals differ in this aspect? A step in this direction is given in Fig 4B which correlates the magnitude of spiking and high gamma LFP responses, but does not dive into the tuning properties. This comparison should be discussed and/or developed more.

REPLY: In Figure 2, the Multi-Unit Activity (MUA) and Local Field Potential (LFP) examples are not obtained from the same channels; they represent different recording sites. This has been clarified on p. 4.

To explore differences in tuning properties in more detail and in response to comment 3 of reviewer 1, we conducted an analysis of the tuning of Multi-Unit Activity (MUA), High Gamma, and Local Field Potential (LFP) Event-Related Potentials (ERPs). We have added a paragraph on p. 7.

15) In Figure 3, are the MUA and LFP sites individually ordered by z-score or is the shown topography the same? It would be nice if the relation between MUA and LFP would be observable. The colored brackets are difficult to see unless zoomed in quite far.

REPLY: No, the Multi-Unit Activity (MUA) and Local Field Potential (LFP) sites are individually ordered, and the topography is not the same. To improve the clarity of the figure, we have made adjustments.

Instead of using brackets, we divided the plot into distinct blocks, each representing a group of channels. We outlined each block to indicate the group, enhancing the visual separation between groups. Below is the updated version of the figure.

16) The interactions between shape and category appear to be pronounced (large proportion of the responsive cells). This makes the interpretation a bit complicated.

REPLY: This is correct. We have elaborated on these interactions on p. 11 of the discussion.

17) Figure 4 lacks y-axis labels. Please add them.

REPLY: The y – axis labels have been added.

18) The dissimilarity analysis should be unpacked. I found it difficult to follow and relate to the other analyses.

REPLY: We added an explanation in the Results section, p. 7

19) There seem to be some remarkable differences between the time-based decoding results for MUA and LFP that are now summarized as LFP giving lower accuracy. Actually, the very sustained decoding accuracy of array2-MUA seems more transient in the LFP, while for array1 the LFP seems more sustained than the MUA, and for array4 the high decoding in MUA seems almost absent in LFP. I think there's more happening here, than just weaker accuracy. The discussions of these differences should be expanded. (Fig 7 vs fig S6)

REPLY: For unknown reasons, the LFP signal of array 4 was of bad quality, which explains the difference with the MUA decoding and also makes it difficult to draw firm conclusions about potential differences between MUA and high - gamma decoding. We felt that a detailed comparison between the temporal dynamics of MUA vs high - gamma decoding was beyond the scope of this study.

20) In the discussion, the authors mention the spatial scope of voxels, arrays and neurons in relation to what is known about shape/category coding and what is shown here. I think this is an important point that goes back to trying to bridge human fMRI and monkey ephys. Based on earlier studies, what would we expect in terms of single site signals and array populations?

REPLY: A single fMRI voxel of 8 mm³ contains approximately 400,000 neurons, whereas the Utah array samples from a 4 by 4 mm patch of cortex. Assuming each electrode samples a 200 micron area the cortical volume covered by one array would be 3.2 mm³ and the volume sampled by a single electrode would be 0.004 mm³. Only in the case of a very homogeneous population of neurons (such as the face-selective neurons in a macaque face patch, or the face neurons in Decramer et al., 2021⁵) would we expect to see effects on an array similar to an fMRI voxel. In our experiments, we observed more heterogeneous populations of neurons which were primarily driven by shape type. This has been added to the discussion on p. 11.

21) The authors conclude that: Together, these findings highlight the complexity of neural mechanisms underlying object processing and the importance of using multiple techniques to uncover these representations. While the population as a whole showed strong shape tuning and only very limited category selectivity, we found a large neuronal diversity and distinct interactions between shape and category at the single-channel level in human LOC.

Important control analyses are missing that are needed before this claim can be accepted. I'm not sure if I completely agree with that and I think a scheme showing a proposed (hierarchical) shape/category coding would help clarify this point. "The population as a whole" here seems to refer to the collection of single sites or the average of those. The authors themselves show and argue that the population as a whole (in the sense of the 'the dynamic non-averaged population activity) does in fact have decodable category information.

REPLY: We apologize for the confusion. We have replaced 'population as a whole' with 'the individual recording sites' to clarify this sentence. It is indeed correct that the decoding analysis showed category information, so we added the following sentence on p. 12: "While individual recording sites showed strong shape tuning and only very limited category selectivity, we found a large neuronal diversity and distinct interactions between shape and category at the single-channel level in human LOC, whereas the populations of neurons showed significant decodable category information."

22) Line 413: "The fixation point included the average receptive field of the MUA for each array." What does it mean that the fixation point includes a receptive field? Do you mean that the receptive field included the fixation point? Was that true for all receptive fields?

REPLY: This is correct. Since we only analyzed data from visually responsive sites, the stimuli we presented at the fixation point appeared in the receptive fields of the neurons. We have rephrased this sentence on p. 14.

23) Line 456: "For the high – gamma responses, due to lower Signal to Noise Ratio, we performed the 1 – way Anova between the baseline and the post – stimulus interval only for the two most preferred conditions per channel." This is statistically incorrect. The ANOVA assumes independently generated data, it is not correct to first make a selection (e.g. max) on the test statistic and to then do an ANOVA.

REPLY: Following the reviewer's correction, we identified visually responsive sites using all conditions, not just the two most preferred ones. We observed that all channels in all four arrays were visually responsive, likely due to the low signal-to-noise ratio (SNR). We subsequently replicated the entire analysis using all channels and obtained consistent results at both the single-channel and population

levels. The only exception was noted in the dissimilarity analysis for array 4, where the correlation between the neural dissimilarity matrix and the shape dissimilarity matrix did not reach significance with this approach.

Single Channel Analysis:

Dissimilarity analysis:

A.

B.

Decoding:

DNNs:

24) Legend of Fig. 2. What is the meaning of “channel ½”? These channels are not visible in the figure itself.

REPLY: This has been corrected. It is now ‘channel in A, channel in B, and channel in C’.

25) Legend of Fig. 7B. Upper/lower panel should be changed into left/right panel.

REPLY: This has been corrected.

Reviewer #3 (Remarks to the Author):

Bougou et al

Neuronal tuning and population representations of shape and category in human visual cortex

Bougou et al perform exciting recordings of neuronal spiking activity in the human temporal lobe in 3 patients with drug-resistant epilepsy. Remarkably, the patient and the ethical committee allowed the implantation of an invasive Utah array devices, presumably outside the epileptogenic zone (see Patients section), which is extremely rare in patients with epilepsy. In one patient, there were even two Utah arrays implanted! Through these recordings, the investigators provide a characterization of the tuning properties of neurons in the human occipital cortex, an area of which we know essentially nothing about at the neurophysiological level. The authors analyze the data in terms of shape and category tuning, a concept that I have many doubts about. But independently of the complex notion of “category”, what is particularly exciting about this study is the possibility of describing the responses of neurons in occipital cortex to different stimuli.

(1) The implantation of Utah arrays in non-epileptogenic tissue in the human brain is fascinating from a scientific standpoint and opens up a lot of ethical questions. It would be interesting to document any information the authors may have about the degree of damage introduced by the implantation of such large electrode arrays. To my understanding, there is no clinical justification for the implantation of such arrays. Most Institutional Review Boards have approved the implantation of electrodes for clinical reasons in epilepsy patients but have traditionally frowned upon the implantation of Utah arrays, except in a few intraoperative studies or in cases where the Utah array is implanted in tissue to be

resected (Utah arrays have also been implanted in healthy tissue in paraplegic patients for the purpose of building brain-machine interfaces). Given that the data have already been recorded, there is no point in dwelling too much on this. But as a precedent, any information provided by the authors on this would be quite useful for the scientific community on whether this type of studies should be allowed in the future.

REPLY: We included the clinical background information in the Methods section on p.12.

(2) I struggle with the use of the terms “shape” and “category” in this study.

(2a) I do not understand the sense in which the authors argue that shape and category are “dissociated”. In lines 423-430, the authors explain that every “shape” contains stimuli from each “category” and every “category” contains stimuli from each “shape” (Figure 1B). This seems to be a rather loose description of the concepts of shape and category. In ref. 21, the authors specifically designed synthetic stimuli where shape and category were independent. That is, there were stimuli pairs that had a short distance (at least in pixel space) in shape but belonged to different categories. And there were stimuli that were very far (at least in pixel space) and belonged to the same category. Perhaps the stimuli in Fig. 1b loosely “appear” to have similar shapes in some colloquial sense along a column, but this is not a rigorous definition of shape.

REPLY: We thank the reviewer for this suggestion. To quantify the shape type differences in an objective manner, we employed the formula developed by Bao et al. 2020⁶ who suggested that this formula captures the most important shape dimension that structures object space. This formula is based on the parameters of perimeter and area (as also described in Yargholi & Op De Beeck 2023⁷). We applied this formula to calculate the “Aspect Ratio,” which represents a single dimension of shape, particularly the distinction between “stubby” and “spike” shapes (added in Methods). We hypothesized that this Aspect Ratio would correlate with our behavioral shape ratings. Using this quantification, we constructed the “Aspect Ratio” dissimilarity matrix by calculating the pairwise absolute differences in Aspect Ratio between all pairs of stimuli. We calculated the correlation between the behavioral dissimilarity matrices and the Aspect Ratio dissimilarity matrix. The results of this analysis are illustrated in the scatter plots and correlation bar plots in the figure provided below. Our findings reveal that there is a significant correlation between the Aspect Ratio dissimilarity matrix and both the shape and silhouette matrices, with the highest correlation observed with the shape matrix (Figure S10). This demonstrates that our approach of grouping stimuli based on shape is underpinned by a quantitative measure, as previously described in Bao et al. 2020⁶. We have included this quantification of shape type in the Methods section on p. 18.

To provide further insights, we included the "Aspect Ratio" matrix in our dissimilarity analysis and compared it with the neural dissimilarity matrices, as illustrated in the updated version of Figure 4.

For arrays 1 and 2, we observed a significant correlation between the neural dissimilarity matrices and the Aspect Ratio matrix. Notably, this correlation was even more pronounced than with the shape and silhouette matrices, suggesting a link between multichannel neural responses and the quantification of shape based on Aspect Ratio.

In the case of array 3, we also found a significant correlation with the Aspect Ratio, although it was somewhat lower compared to the correlations with shape and silhouette. This indicates that, in array 3, shape and silhouette may have a relatively stronger influence on neural responses.

However, for array 4, our analysis did not reveal a significant correlation between the neural dissimilarity matrix and the Aspect Ratio. This suggests that for this particular array, the neural responses may not be as strongly influenced by Aspect Ratio as they are by other features. These results have been added on p. 8.

(2b) One way in which studies have tried to separate categories from shape is by asking whether preferred stimuli for neurons can be better described based on category tuning or shape tuning and the answer in the inferior temporal cortex invariably is shape tuning (see studies of Earl Miller in ref. 21), without any evidence of categorical tuning. A recent example is Bardon et al PNAS 2022. Given that the current occipital locations seem to be well before inferior temporal cortex in the visual hierarchy, it would be remarkable to find any categorical tuning and no evidence for categorical tuning has been reported in monkeys in occipital or temporal cortex.

REPLY: This is indeed a very important point. First of all, at this moment it is unclear whether the area we recorded from in humans (LOC) is homologous to the macaque TEO, posterior TE or anterior TE. There is some (partially unpublished) evidence that LOC might correspond more to posterior TE but more research is necessary. Secondly, one cannot simply assume that the properties of macaque inferotemporal neurons are identical to those of human LOC neurons. It is possible that the human ventral visual pathway already has an explicit representation of object categories which does not exist in macaques because animals have to learn these categories in adulthood through extensive training whereas for humans these categories are very familiar. In fact, the fMRI study of Bracci et al. 2016² reported interactions between shape type and category in the human LOC, but without neural recordings it was impossible to determine what these interactions really mean. Our findings indicate that LOC neurons share a striking similarity with macaque inferotemporal neurons, and that category information is present at the population level. We have added this reasoning in the discussion on p. 11.

(2c) Another way in which studies have tried to compare image features is by using neural network models and asking whether a model can explain neuronal responses (e.g. Yamins and DiCarlo PNAS 2014). Again, the conclusion from these studies is that activity in inferior temporal cortex is well explained by shape tuning without any categorical information. I would suggest removing all the discussion of shape and categories in the current study.

REPLY: The motivation for this study came from a previous fMRI study (Bracci et al. 2016²) that showed shape type – category interactions in the human lateral occipital cortex, which raised the possibility that this region in humans may exhibit category representations unlike the macaque inferotemporal cortex. We argue in the discussion on p. 11 that one should not assume that all properties of neurons in the human lateral occipital cortex are identical to those of neurons in the macaque inferior temporal cortex. Moreover, object categories in macaques require extensive training whereas object categories in humans are known since childhood, and macaques may never learn object categories at the semantic level. Our results clearly show primarily shape-based representations although object category can be decoded from the population of neurons.

We have added an analysis based on deep neural networks on p. 10 (see also reply to comment 2 of reviewer 2), which shows that the highest correlation with our neural data occurs in the intermediate layers. Moreover, our quantification of shape type added on p.8 provides strong support for the validity of the distinction between shape type and category in our study.

3)3a. It would be very nice to show individual examples of raster plots to evaluate the quality of the data.

REPLY: The figures below (added now as Supplementary Figure 2) illustrate raster plots of the Single-Unit Activity (SUA) from the example channels per array. The rows in each plot represent trials, while the columns represent the different shapes or categories.

Example channels SUA:

3b. What are the units for the activity fig. 1a? Spikes/second? If so, why are the firing rates so low? Recordings in the macaque occipital cortex tend to show that individual neurons can fire tens if not hundreds of spikes/sec.

REPLY: This is correct, but in this figure we showed the average response of all visually responsive channels to the LOC localizer stimuli (which were most likely not optimal for the neurons we recorded from) to illustrate the effect of scrambling, without any attempt to optimize the stimulus to the recorded neurons (as is frequently done in macaque studies). Note also that the responses of the example channels in Figure 2 reach 30 to 100 spikes/sec.

3c. Is Fig. 1A showing the average of all the channels in the array? If so, why is this interesting? Perhaps the issue here is the average over neurons with very different stimulus properties. The notion of LOC and “localizer” derived from fMRI studies is not very useful or relevant with real neurophysiological data. This type of averaging seems to occlude the interesting responses in the data. In fact, even though Fig. 2 does not show units either, the numbers on the y-axis are much larger than those in Fig. 1, suggesting that the averaging is at least partly probably to blame for the low activity in Fig. 1.

REPLY: We agree with this comment. We now simply state that all arrays showed a significant effect of image scrambling to link our findings to the existing human fMRI literature (the original J Neurosci paper by Kourtzi and Kanwisher ⁸ has been cited more than 800 times, a follow-up review paper by Grill-Spector, Kourtzi and Kanwisher ⁹ has been cited more than 1600 times).

(4) Continuing with the challenge of defining shape and category with uncontrolled stimuli, it is difficult to interpret the category and shape tuning results in Fig. 2. I would find it much more instructive to show the responses of individual channels (individual neurons if the authors can do spike sorting, depending on the quality of the data) to all the stimuli and report the degree of selectivity across all the stimuli without making any distinction along the arbitrary dimensions of category and shape. I would even use the term “shape selective” in this analysis to refer to neurons that show different responses among all 54 stimuli.

REPLY: We agree with the reviewer that studying shape selectivity of neurons in this area is very interesting and we intend to write a separate paper about these characteristics using different stimulus sets. However, the stimulus set we used in the current study was not designed to study shape selectivity as such, but rather to investigate shape and category representations motivated by the fMRI study of Bracci et al. 2016². To illustrate shape selectivity, we calculated d' primes for all responsive channels based on the best and worst net response across all 54 stimuli. It is clear from the figure below that we frequently measured very high d' values ($d' > 2$) for all four arrays. This figure has been added as Supplementary Figure 3.

(5) It is hard to assess the quality of the MUA data because the authors do not show any waveforms of action potentials, which is customary to do in the field. It would also be useful to do spike sorting to separate the activity of individual neurons.

REPLY: We have included spike wave forms for the example channels to illustrate the quality of the recordings (Figure S2). We also spike sorted the example channels and confirmed their selectivity.

Spike sorting inevitably favors the large action potentials, therefore we prefer to report the MUA in this study to not bias the data in the direction of large pyramidal neurons while ignoring the contribution of smaller neurons.

(6) Many investigators using Utah arrays report that it is hard to discern spiking activity immediately and they have to wait days/weeks/sometimes even months to get high spiking activity quality. Given the paucity of Utah array data from the human brain for ethical reasons, it would be useful if the authors could document the quality of the data and the evolution of quality over time during the period of 14 days that the patients stayed in the hospital. At a bare minimum, it would be useful to report when the recording in the paper took place and to show quantification of the actual waveforms.

REPLY: This is correct for Utah array implantations in macaques. However, due to small adaptations in the insertion technique, we usually can record spiking activity in humans on the day after surgery (in macaques we use the same technique and typically see spikes after one week). The signal frequently improves over time in the 2-week period that the patients are in the hospital. In the methods, we have added information about which day after implantation the data were recorded on p. 14. Since the recordings reported in this study took place on a single day, we cannot demonstrate the quality of the recordings over time. However, in a second study focused on tolerance to stimulus transformations we recorded with images of shapes in the first days after implantation. We will document the evolution of the signal quality in this second paper.

(7) Given the paucity of data from human occipital cortex neurons and the exciting recording performed here, plus the possibility that it will be very hard to get data from these areas in the human brain, it would be amazing to document these responses in much more detail. What is the latency of the responses? What is the size and shape of their receptive fields? How come all/most of the receptive fields seem to be centered in the fovea (this seems counterintuitive based on what we know from similar areas in monkeys)? Do nearby channels (neurons) share similar tuning properties (their last figure hints at this)?

REPLY: We calculated the response latencies for all visually responsive channels (MUA and high gamma) and added this information as Supplementary Figure 4 and in the Methods section. Due to time constraints, we could not perform a detailed receptive field mapping with the current stimulus set. With chronically implanted multielectrode arrays, presenting 54 stimuli at 100 positions with 8 trials per condition (shape x location combination) would require 43,200 trials, which is impossible to achieve. Even a coarse mapping with 25 positions and rapid stimulus presentation (3 per second) would require 54 minutes of passive fixation, which is more than most patients can do. However, to give an idea about receptive field size and location, in the same figure, we also show example receptive fields (RFs) for all arrays recorded with a different stimulus set. To map the receptive field (RF) of arrays 1,2, and 4, we showed a 4 deg flickering checkerboard (20 Hz) at 25 positions on the screen, covering a 10 x 10 degree area for arrays 1 and 2 and a 14 x 14 degree area of the visual field for array 4. In the case of array 3, instead of the checkerboard, we presented a 1 deg image of a body at 25 different positions, covering a 14 x 14 degree area of the visual field. Note that Decramer et al. 2019 ⁴ also reported response latencies and receptive field sizes in two other patients implanted with Utah arrays.

References:

1. Julian, J. B., Fedorenko, E., Webster, J. & Kanwisher, N. An algorithmic method for functionally defining regions of interest in the ventral visual pathway. (2012) doi:10.1016/j.neuroimage.2012.02.055.
2. Bracci, S. & Op de Beeck, H. Dissociations and Associations between Shape and Category Representations in the Two Visual Pathways. *J. Neurosci.* **36**, 432–444 (2016).
3. Kriegeskorte, N. *et al.* Matching Categorical Object Representations in Inferior Temporal Cortex of Man and Monkey. *Neuron* **60**, 1126–1141 (2008).
4. Decramer, T. *et al.* Single-cell selectivity and functional architecture of human lateral occipital complex. *PLoS Biol.* **17**, (2019).
5. Decramer, T. *et al.* Single-Unit Recordings Reveal the Selectivity of a Human Face Area. *J. Neurosci.* **41**, 9340–9349 (2021).
6. Bao, P., She, L., McGill, M. & Tsao, D. Y. A map of object space in primate inferotemporal cortex. *Nature* **583**, 103 (2020).
7. Yargholi, E. & de Beeck, H. O. Category Trumps Shape as an Organizational Principle of Object Space in the Human Occipitotemporal Cortex. *J. Neurosci.* **43**, 2960–2972 (2023).
8. Kourtzi, Z. & Kanwisher, N. Cortical regions involved in perceiving object shape. *J. Neurosci.* **20**, 3310–3318 (2000).
9. Grill-Spector, K., Kourtzi, Z. & Kanwisher, N. The lateral occipital complex and its role in object recognition. *Vision Res.* **41**, 1409–1422 (2001).

REVIEWER COMMENTS

Reviewer #1 (Remarks to the Author):

The authors did an excellent job in addressing my concerns.

Reviewer #2 (Remarks to the Author):

2nd Review of

Neuronal tuning and population representations of shape and category in human visual cortex

The authors have addressed most of my concerns in an excellent revision. I appreciated the inclusion of modeling with the convolutional networks.

A few relatively minor points remain, which the authors should be able to address.

1) The reader would appreciate reading in the results section what the patients were doing. I.e. please indicate that they were awake and fixating. Also indicate at the beginning of results how long the arrays stayed in place before they were explanted.

2) Maybe I missed it, but I found no information about the size of the stimuli in deg.

3) S_{width} : please explain in results, when this measure is described that it ranges from 0 for neurons that give an equal response to all stimuli to 1 for neurons that only respond to one of the stimuli and not to any of the others.

4) Typo: line 242 iteration -> iterations

5) Legend Fig. 8: the legend describes panel A-C, but the figure also has a panel D.

6) Figure S5B The legends indicates that there is a panel on the right that shows High Gamma but that panel is not there.

Reviewer #3 (Remarks to the Author):

Bougou et al revision

(1) The authors did not address the question.

Their response was:

"We included the clinical background information in the Methods section on p.12"

First, I assume that they meant p. 13.

I repeat the critical aspects of the question that were not addressed.

The implantation of Utah arrays in non-epileptogenic tissue in the human brain is fascinating from a scientific standpoint and opens up a lot of ethical questions. It would be interesting to document any information the authors may have about the degree of damage introduced by the implantation of such large electrode arrays. To my understanding, there is no clinical justification for the implantation of such arrays. Most Institutional Review Boards have approved the implantation of electrodes for clinical reasons in epilepsy patients but have traditionally frowned upon the implantation of Utah arrays, except in a few intraoperative studies or in cases where the Utah array is implanted in tissue to be resected (Utah arrays have also been implanted in healthy tissue in paraplegic patients for the purpose of building brain-machine interfaces). Given that the data have already been recorded, there is no point in dwelling too much on this. But as a precedent,

any information provided by the authors on this would be quite useful for the scientific community on whether this type of studies should be allowed in the future.

(2) The definition of shape and category remain ill-posed.

(2a1) The authors introduce the words "stubby" and "spiky", which make the definitions even more complex. The field has moved away from using words like shape, category, stubby, etc. and now focuses on using computational models to define visual features.

(2a2) The authors introduce a figure with presumed correlations. Unfortunately, part A of this figure follows very poor standards, with a color scale that does not have numbers and is therefore uninterpretable. Part B of this figure is also not very clear (what exactly is a point in this figure?). Even if one were to accept the aspect ratio as a useful metric (which is very unclear), the correlation with the concept of "shape" is minimal ($R=0.22$, meaning that the explained variance between these two variables is on the order of 4%!).

(2a3) As a lesser concern, these weak correlations are also different for different arrays (but this is fine because the arrays may be in different locations)

(2b) The authors insist that their results demonstrate evidence of categorical tuning, but this is not the case. I agree that one should not assume that the macaque inferior temporal cortex is identical to human visual cortex. But one should also not assume that neurons in human visual cortex represent categorical information. To really demonstrate categorical tuning, it would be important to design stimuli that explicitly dissociate category and visual shapes. As noted in the original review, this is what Miller and colleagues did. The current stimuli do not allow to draw conclusions about categorical tuning. The analyses with neural networks further confirm that the neurons in the current study represent visual shapes and not categorical information.

I suggest that the authors remove the discussion of shape and category and instead focus on how neurons in human visual cortex represent information. The paper would be stronger without the ill-defined notions of shape and category.

(3a) The quality of the new raster plot figures is very poor, making it very hard to appreciate the responses. The figures are extremely blurry.

It is nice that the waveforms have a scale bar, but this is completely unreadable due to blurring. Which of the example units do the authors believe carries any categorical information? Through the blurring, the examples seem to suggest the encoding of features in some images and not others within a category, if I am interpreting the blurry rasters correctly.

(3c) The authors insist on the notion of scrambling images, which is not very useful. Intriguingly, instead of providing a scientific justification, the authors mention the number of citations in some papers. Scrambling destroys all kinds of image features, scrambling may change the level of attention in the images, scrambling may lead to different microsaccades and saccades, the list of confounds is long.

The authors provided interesting responses, especially response 6 about the evolution of spiking activity post implantation.

In sum, these are very valuable and exciting recordings from the human occipital cortex. We know almost nothing about the human occipital cortex and most of the recordings from the human brain have focused on the temporal lobe, medial temporal lobe, suprachiasmatic nuclei in Parkinson's disease, motor cortical areas in paraplegic patients. The current study provides a unique set of data. The discussion of shape and category seems misguided and does not help interpret the responses. But the data are extremely valuable and worth reporting.

POINT TO POINT

Reviewer #1 (Remarks to the Author):

The authors did an excellent job in addressing my concerns.

Reviewer #2 (Remarks to the Author):

The authors have addressed most of my concerns in an excellent revision. I appreciated the inclusion of modeling with the convolutional networks.

A few relatively minor points remain, which the authors should be able to address.

1) The reader would appreciate reading in the results section what the patients were doing. I.e. please indicate that they were awake and fixating. Also indicate at the beginning of results how long the arrays stayed in place before they were explanted.

REPLY: We included details about the patients' tasks during the experiment at the beginning of the results section on page 4. The information regarding the duration of the arrays' implantation, was already stated in the same paragraph as "After 14 days, the arrays and grids were removed without any further incisions."

2) Maybe I missed it, but I found no information about the size of the stimuli in deg.

REPLY: We apologize for the oversight. The stimuli used in our study had an approximate diameter of 8 visual degrees. Each stimulus measured 400 x 400 pixels, with approximately 50 pixels dedicated to the background. Therefore, each pixel represented approximately 0.026 degrees of visual angle. This information has been added in the Methods section, page 15.

3) S_{width} : please explain in results, when this measure is described that it ranges from 0 for neurons that give an equal response to all stimuli to 1 for neurons that only respond to one of the stimuli and not to any of the others.

REPLY: We included this explanation in the results section, page 5.

4) Typo: line 242 iteration -> iterations

REPLY: The typo has been corrected.

5) Legend Fig. 8: the legend describes panel A-C, but the figure also has a panel D.

REPLY: The legend has been corrected.

6) Figure S5B The legends indicates that there is a panel on the right that shows High Gamma but that panel is not there.

REPLY: The reviewer likely refers to Figure S4B, which indeed mistakenly refers to a High Gamma panel. The legend has been corrected.

Reviewer #3 (Remarks to the Author):

1) The authors did not address the question.

Their response was:

“We included the clinical background information in the Methods section on p.12”

First, I assume that they meant p. 13.

I repeat the critical aspects of the question that were not addressed.

The implantation of Utah arrays in non-epileptogenic tissue in the human brain is fascinating from a scientific standpoint and opens up a lot of ethical questions. It would be interesting to document any information the authors may have about the degree of damage introduced by the implantation of such large electrode arrays. To my understanding, there is no clinical justification for the implantation of such arrays. Most Institutional Review Boards have approved the implantation of electrodes for clinical reasons in epilepsy patients but have traditionally frowned upon the implantation of Utah arrays, except in a few intraoperative studies or in cases where the Utah array is implanted in tissue to be resected (Utah arrays have also been implanted in healthy tissue in paraplegic patients for the purpose of building brain-machine interfaces). Given that the data have already been recorded, there is no point in dwelling too much on this. But as a precedent, any information provided by the authors on this would be quite useful for the scientific community on whether this type of studies should be allowed in the future.

REPLY: Indeed, we meant to reference page 13 instead of 12; we apologize for any confusion. As stated on page 13, there is a clinical justification for the implantation of the arrays, which is our clinical study focusing on the microscale dynamics of epileptic networks. We have previously presented a poster related to this study at EANS 2023 in Barcelona titled “Microscale Dynamics of Epileptic Networks: Insights from Multiunit Activity Analysis in Neurosurgical Patients with Refractory Epilepsy.” Additionally, we are working on making a dataset of seizure data recorded from 5 patients implanted with Utah arrays (MUA and LFP) along with SEEG available to the scientific community. The dataset, titled “Mesoscale Insights in Epileptic Networks: A Multimodal Intracranial Dataset,” has already been submitted to the EBRAINS repository and is currently undergoing curation. Furthermore, we have drafted a manuscript for submission to the Journal of Scientific Data, describing the dataset.

We acknowledge the importance of documenting the potential degree of damage introduced by the implantation procedure. As such, we have included this information in the Methods section on pages 13 - 14 and in Supplementary Figure 18.

2) The definition of shape and category remain ill-posed.

(2a1) The authors introduce the words “stubby” and “spiky”, which make the definitions even more complex. The field has moved away from using words like shape, category, stubby, etc. and now focuses on using computational models to define visual features.

REPLY: It's important to clarify that we employ these terms to maintain consistency with the nomenclature used in several prior publications where they have been utilized to describe the stimulus set. Moreover, the use of these terms is widespread across literature related to monkey IT and the human ventral stream, where they are commonly used to describe the properties of neurons and identify hierarchical levels.

While advancements in deep neural networks (DNNs) have allowed for the examination of more detailed properties of objects and the exploration of visual features, this does not negate the relevance of the terminology of shapes and semantic categories. Notably, previous research, including studies by Zeman et al. (2020) and Kubilius et al. (2016), has explicitly demonstrated separate tuning for the

dimensions of shape and category using computational models. Furthermore, computational models such as convolutional neural networks (CNNs) trained on image recognition also refer to these terms. For example, the fully connected layers of CNNs are used for classifying input images into labels, which involves a categorization process assigning a semantic category to an object.

We have added a sentence explaining why we use these terms in the introduction on p. 3, and we have adjusted our use of this terminology in the Discussion (see also our response to comment 2b).

(2a2) The authors introduce a figure with presumed correlations. Unfortunately, part A of this figure follows very poor standards, with a color scale that does not have numbers and is therefore uninterpretable. Part B of this figure is also not very clear (what exactly is a point in this figure?). Even if one were to accept the aspect ratio as a useful metric (which is very unclear), the correlation with the concept of “shape” is minimal ($R=0.22$, meaning that the explained variance between these two variables is on the order of 4%!

REPLY: We apologize for the oversight in not including the color scale in the A panel of the figure. Our intention with this figure was to demonstrate the correlation between the aspect ratio dissimilarity matrix (RDM) and the behavioral and silhouette RDMs, thereby introducing this measure to our analysis and explaining which dimension of the object it captures. We found significant correlations between the aspect ratio RDM and the shape and silhouette RDMs, but not with the semantic category RDM. It's important to note that while the correlation is statistically significant, we do not claim that the aspect ratio is equivalent to the shape RDM. Instead, it represents a measure that aligns with the shape dimension, as described on pages 7 – 8 and in the definition of this measure. Given that aspect ratio is designed to capture a single dimension of a shape, a moderate correlation with the shape RDM is expected. Regarding Panel B, the scatter plots depict the relationship between the dissimilarities of all pairs of stimuli in one RDM with another RDM. We included these scatter plots in response to a comment from the previous revision (Reviewer 1, Point 4), which suggested that barplots of correlation may compress the results and that a scatter plot would provide more detailed insight.

(2a3) As a lesser concern, these weak correlations are also different for different arrays (but this is fine because the arrays may be in different locations)

REPLY: We acknowledge that such differences can be expected, especially if the arrays are positioned in different locations. Additionally, we addressed the issue of low correlation values in the previous revision (Reviewer 1, Point 4).

(2b) The authors insist that their results demonstrate evidence of categorical tuning, but this is not the case. I agree that one should not assume that the macaque inferior temporal cortex is identical to human visual cortex. But one should also not assume that neurons in human visual cortex represent categorical information. To really demonstrate categorical tuning, it would be important to design stimuli that explicitly dissociate category and visual shapes. As noted in the original review, this is what Miller and colleagues did. The current stimuli do not allow to draw conclusions about categorical tuning. The analyses with neural networks further confirm that the neurons in the current study represent visual shapes and not categorical information.

I suggest that the authors remove the discussion of shape and category and instead focus on how neurons in human visual cortex represent information. The paper would be stronger without the ill-defined notions of shape and category.

REPLY: We have removed all references to ‘category tuning’ from the manuscript: we only show that a linear decoder can reliably classify category information from the responses, but we do not imply in any way the presence of categorical tuning. We would like to point out that the Miller study was done in monkeys that had to discriminate between learned categories (on page 3), whereas we used categories that the subjects knew since childhood. We acknowledge that the term ‘shape selectivity’ may have been too strong since neurons could be responding to shape, texture, shading or other

stimulus features. Therefore, we have replaced this in the discussion with ‘visual selectivity’ (on pages 10, and 12). See also our response to comment 2a1.

(3a) The quality of the new raster plot figures is very poor, making it very hard to appreciate the responses. The figures are extremely blurry.

It is nice that the waveforms have a scale bar, but this is completely unreadable due to blurring.

Which of the example units do the authors believe carries any categorical information? Through the blurring, the examples seem to suggest the encoding of features in some images and not others within a category, if I am interpreting the blurry rasters correctly.

REPLY: We apologize for any blurriness observed in the figure. All our figures are saved in high resolution, so this issue may have arisen during the creation of the PDF from the journal's online platform. We have included the figure below for your reference and will take steps to address any blurriness. Additionally, we have increased the size of the scale bar to ensure readability. The raster plots depict the activity of single units recorded from all arrays for the different shapes, and they also illustrate the activity of a single unit recorded from array 3 across different categories. From the latter, it is evident that the neuron exhibits stronger responses to the "animals" category compared to others, indicating that it carries categorical information.

(3c) The authors insist on the notion of scrambling images, which is not very useful. Intriguingly, instead of providing a scientific justification, the authors mention the number of citations in some papers. Scrambling destroys all kinds of image features, scrambling may change the level of attention in the images, scrambling may lead to different microsaccades and saccades, the list of confounds is long.

REPLY: We acknowledge the concerns raised regarding the scrambling method utilized in our study. We agree that this method has limitations and introduces various confounds. As mentioned in our previous revision, we do not draw any direct conclusions from this experiment. Instead, we use it to establish a link between our findings and previous fMRI results using the LOC localizer.

In Figure S1, we illustrate the positioning of our arrays in relation to the LOC parcels as defined in Julian et al. 2012. These parcels were identified using scrambled vs. non-scrambled stimuli. Therefore, we believe it is valuable to present the results from our neural recordings using a similar experimental setup. We view the extensive literature spanning over 20 years, which has consistently employed this experiment to identify the LOC region in human fMRI studies, as a scientific justification for replicating this experiment. It is important to emphasize that we do not draw definitive conclusions from our results but rather compare our findings to this well-established literature from human fMRI studies.

The authors provided interesting responses, especially response 6 about the evolution of spiking activity post implantation.

In sum, these are very valuable and exciting recording from the human occipital cortex. We know almost nothing about the human occipital cortex and most of the recordings from the human brain have focused on the temporal lobe, medial temporal lobe, suprachiasmatic nuclei in Parkinson's disease, motor cortical areas in paraplegic patients. The current study provides a unique set of data. The discussion of shape and category seems misguided and does not help interpret the responses. But the data are extremely value and worth reporting.